# Log-Polar Space Convolution Layers

**Bing Su, Ji-Rong Wen**[*]
Beijing Key Laboratory of Big Data Management and Analysis Methods
Gaoling School of Artificial Intelligence, Renmin University of China
Beijing 100872, China
subingats@gmail.com; jrwen@ruc.edu.cn

## Abstract

Convolutional neural networks use regular quadrilateral convolution kernels to extract features. Since the number of parameters increases quadratically with the size of the convolution kernel, many popular models use small convolution kernels, resulting in small local receptive fields in lower layers. This paper proposes a novel log-polar space convolution (LPSC) layer, where the convolution kernel is elliptical and adaptively divides its local receptive field into different regions according to the relative directions and logarithmic distances. The local receptive field grows exponentially with the number of distance levels. Therefore, the proposed LPSC not only naturally encodes local spatial structures, but also greatly increases the single-layer receptive field while maintaining the number of parameters. We show that LPSC can be implemented with conventional convolution via log-polar space pooling and can be applied in any network architecture to replace conventional convolutions. Experiments on different tasks and datasets demonstrate the effectiveness of the proposed LPSC.

## 1   Introduction

Convolutional neural networks [1, 2] have achieved great success in the field of computer vision. The size of the convolution kernel determines the locally weighted range of the image or feature map, which is called the *local receptive field (LRF)*. In many computer vision tasks such as image classification [2, 3, 4] and intensive prediction [5, 6, 7], larger LRF is generally desired to capture the dependencies between long-distance spatial positions and a wide range of context information. Simply increasing the size of the convolution kernel is not plausible because the number of parameters increases quadratically with the size.

In practice, commonly used techniques to obtain larger receptive fields include adding pooling layers, replacing a single-layer large convolution kernel with multi-layer small convolution kernels, and using dilated convolutions [8, 9]. The pooling process often causes information loss. Increasing the number of convolutional layers may cause vanishing gradients and make training more difficult. Moreover, going deeper with small kernels may not indicate a larger receptive field. A plain CNN with all $3 \times 3$ convolution kernels cannot be too deep without residual connections. Some studies [10] have found that ResNets behave like ensembles of shallow networks. Regardless of the actual depth, the effective number of layers for ResNets maybe limited. That is, even if a ResNet with hundreds of layers is stacked, its actual receptive field may be equivalent to that of a shallow network.

According to the effective receptive field (ERF) theory [11], the ERF is proportional to the square root of the depth and directly proportional to the kernel size. Therefore, it is easier to achieve a large ERF by increasing the kernel size than by adding layers. The success of Vision Transformers [12, 13] may also reveal the effectiveness of large local windows, while various sparse attention mechanisms [14,

---

[*]Corresponding author: Ji-Rong Wen.

36th Conference on Neural Information Processing Systems (NeurIPS 2022).

15, 16] for Transformers are proposed to allow larger LRFs with limited increases of calculations. In this paper, we reconsider lightweight CNNs with large convolution kernels. Dilated convolution kernels are able to increase the LRFs greatly, but they are not continuous since not all pixels in the LRF are involved in convolution calculation. The skipped pixels are regularly selected. With the same number of parameters, the larger the LRF, the more pixels are skipped, which may miss some details and cause discontinuity of information.

In addition, conventional and dilated convolutions use regular square kernels. Each position is assigned a different weight within the LRF. All positions are equally treated regardless of the size of the kernel. However, intuitively, the correlation between neighboring pixels and the center pixel is usually higher, while the farther the pixel, the smaller the impact on the center pixel, which is evidenced by statistics from natural images presented in Appendix A.1. The effects of two adjacent pixels that are far away from the center are usually similar, thus they can share the same parameter rather than be assigned different weights separately. As shown in red in Fig. 1(a), according to the configuration of surrounding regions, it can be inferred that the center position is located on the upper edge of the nose. Pixels in the same upper-left outer half-fan-shaped region show that the far upper left of the center point is white fur, but there is little difference in the effects of two specific fur points.

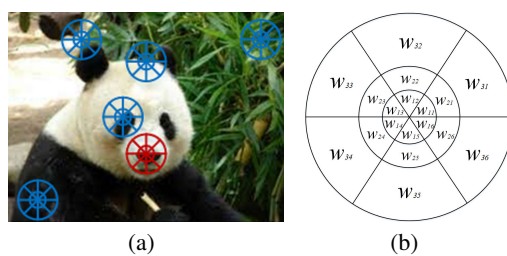

(a)          (b)

Figure 1: (a) At different locations of an image, local contextual pixels can be divided into different regions according to their relative distances and directions in the log-polar space. For each location, pixels falling in the same region are generally similar and can share the same weight. (b) The LPSC kernel. For this example, $L_r = 3, L_\theta = 6, g = 2$, thus there are only 18+1 parameters.

In this paper, we propose a novel *log-polar space convolution (LPSC)* method. The shape of the LPSC kernel is not a regular square, but an ellipse. Parameters of the kernel are not evenly distributed in the LRF, but are assigned in the log-polar coordinate space. As shown in Fig. 1(b), the LPSC kernel divides the LRF into different regions, where regions become larger with the increase of the distance to the center. Pixels that fall into the same region share the same weight. In this way, LPSC can increase the LRF exponentially without increasing the number of parameters. Besides, LPSC naturally imposes a contextual structure on the local neighboring distribution.

The main contributions of this paper include: 1. We propose a new convolution method where the kernel lies in the log-polar space to capture the structured context information and greatly expand the LRF without increasing the number of parameters. 2. We propose log-polar space pooling to up-sample the feature map, by which conventional convolution can be conveniently used to achieve LPSC. 3. We apply LPSC to replace the conventional and dilated convolution in different network architectures including AlexNet, VGGNet, ResNet, DeepLabv3+, and CE-Net. We demonstrate the effectiveness of LPSC through empirical evaluations on different tasks and datasets.

## 2 Related work

**Context pooling.** Our method is highly motivated by shape context [17, 18]. Centered at a reference point, all other points are divided into bins that are uniformly distributed in the log-polar space. The histogram among these bins is used as the descriptor. The statistics in the log-polar space have also been shown to be effective for word recognition in [19]. Geometric blur [20] sparsely samples and aggregates a blurred signal in the log-polar space. Pyramid context [21] pools log-spaced context points at multiple scales. Different from these methods, we design a kernel in the log-polar space for convolution, each region is assigned a weight to aggregate information from the bins. We incorporate the kernel into deep neural networks.

**Methods to increase LRFs.** In [22] and [23], it is found that imposing a regularization on large convolution kernels is equivalent to the superposition of multiple convolution layers with smaller kernels. Based on this observation, many state-of-the-art network architectures use multi-layer small kernels. However, deeper layers may cause vanishing gradients, making the network more difficult to

train. Moreover, according to [11], the effective receptive field (ERF) is proportional to the square root of the depth and proportional to the kernel size. Thus it is easier to achieve a large ERF by increasing the kernel size than by adding layers. We provide a way to increase the LRF without increasing either the number of layers or the number of parameters. In cases where large input or LRF is required but very deep networks are not allowed restricted by resources, our method may be applied to construct a lightweight model.

In [8, 9], atrous (or dilated) convolution increases the LRF by inserting holes (zeros) between parameters in the kernel, where the interval is determined by a dilation rate. Dilated convolution has been applied in different tasks [24, 25, 7, 26, 27, 28]. In [29] and [30], scale-adaptive convolution learns adaptive dilation rate with a scale regression layer. Due to the insertion of holes, not all pixels in the LRF are used for calculating the output. In [31] and [32], this problem is alleviated by hybrid dilated convolution and Kronecker convolution that uses the Kronecker product to share parameters.

**Other convolution methods.** Fractionally strided convolution [33, 34] up-samples the input by padding. In [35], a spatial transformer transforms the regular spatial grid into a sampling grid. Active convolution [36] learns the shape of convolution by introducing the convolution unit with position parameters. Deformable convolution and kernels [6, 37] learn additional offsets or perform resampling to augment the sampling locations, thereby adaptively changing the LRF into a polygon. For active and deformable convolutions, the adapted LRF contains holes, the positions and offsets are learned through additional convolutions, which increases the parameters. Deformable kernels [38] resample the original kernel space and adapt it to the deformation of objects. The offsets for kernel positions also need to be learned. Quasi-hexagonal kernels [39], blind-spot kernels [40], asymmetric blocks [41], and circle kernels [42] also have non-regular shapes, but generally they cannot enlarge LRFs without increasing parameters.

Group convolution [2, 43, 44] and separable convolution [45] do not increase the LRF of kernels. Octave convolution [46] decomposes the feature map into high-frequency and low-frequency features. Multi-scale convolution is performed in [47] and [48]. In [49] and [50], stand-alone self-attention is used to replace convolution. The filter in the attention module lies in a regular and square grid. In [51], the polar transformer network generates a log-polar representation of the input by differentiable sampling and interpolation techniques. The polar transform is applied to a single predicted origin location. In contrast, LPSC performs log-polar pooling via binning and can be applied at any location.

**Differences.** For dilated and other advanced convolutions, the kernel is still performed in a regular grid and all parameters are treated equally. Regardless of the distance from the center, the interval or the sharing range of a parameter is the same among different positions. In contrast, the proposed LPSC expands the LRF in the log-polar space, where near and far regions are distinguished in parameter sharing. The farther away from the center, the larger the range of parameter sharing.

## 3 Log-polar space convolution

Let $\boldsymbol{X} \in \mathbb{R}^{H \times W \times C}$ be the input image or feature map, where $H$, $W$, and $C$ are the height, width, and number of channels of $\boldsymbol{X}$, respectively. $\boldsymbol{W} \in \mathbb{R}^{(2M+1) \times (2N+1) \times C}$ is a conventional convolution kernel with a size of $(2M + 1) \times (2N + 1)$. The central parameter of $\boldsymbol{W}$ is indexed by $(0, 0)$, parameters of $\boldsymbol{W}$ lie in a regular grid $\{(-M, -N), (-M, -N + 1), \cdots, (M - 1, N), (M, N)\}$. The convolution operation is performed in the 2D spatial domain across the channels. For a spatial location $(i, j)$, the output of the conventional convolution is calculated as

$$(\boldsymbol{X} * \boldsymbol{W})(i, j) = \sum_{m=-M}^{M} \sum_{n=-N}^{N} (\boldsymbol{X}(i + m, j + n) \cdot \boldsymbol{W}(m, n)) + b, \tag{1}$$

where $b$ is the bias. Strictly, Eq. (1) actually performs cross-correlation. For convolution, $\boldsymbol{W}$ needs to be rotated 180 degrees. However, since we can view the learned $\boldsymbol{W}$ as the rotated kernel, we follow the common practice of CNN to formulate convolution into Eq. (1). Parameters of the kernel are uniformly distributed in the regular grid, thus each pixel of $\boldsymbol{X}$ falling into the field is weighted by a separate parameter, i.e., all positions are equally treated. However, pixels that have different distances and directions from the center may have different impacts, e.g., pixels adjacent to the center should have larger contributions to the output. Pixels in the input image usually change gently, adjacent pixels far away from the center often have similar impacts on the center. Based on these intuitions,

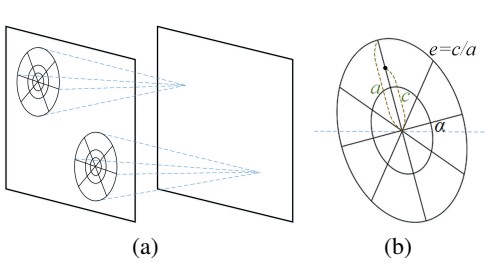

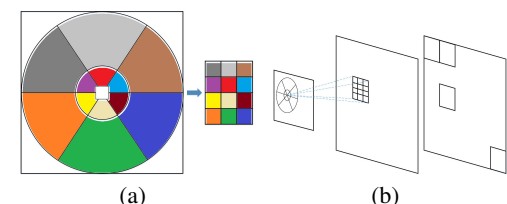

| (a) | (b) |

Figure 2: (a) The LPSC kernel is slid through the feature map. (b) The shape and inclination of the LPSC kernel can be changed.

Figure 3: (a) Log-polar pooling generates a $2L_r \times L_\theta/2$ matrix per position using the pre-computed mask. (b) To perform LPSC on the original input (left), log-polar pooling generates a matrix for each pixel, resulting in an upsampled map (middle), conventional convolution is applied to this map (right).

we design a convolution kernel with a special structure, namely *Log-Polar Space Convolution (LPSC)* kernel, to express a wide range of contextual configurations.

## 3.1 LPSC kernel

As shown in Fig. 1(b), the proposed LPSC kernel lies in the log-polar space and is shaped by the size $2R + 1$, the number of distance levels $L_r$, the number of direction levels $L_\theta$, and the growth rate $g$. The LRF of the kernel is the area of the outermost circle whose radius is $R$. It is uniformly divided into $L_r \times L_\theta$ regions in the log-polar space. Specifically, the log radius is uniformly divided into $L_r$ levels, i.e.,

$$log(R_{l+1}) - log(R_l) = log(R_l) - log(R_{l-1}) = log(g), \qquad (2)$$

where $R_l, l = 1, \cdots, L_r$ is the radius of the $l$-th level and the growth rate $g$ is a hyperparameter controlling the expansion speed. When the center of the kernel is located at position $(c_h, c_w)$, all pixels of $X$ in the range of $\Delta = [c_h - R, c_h + R] \times [c_w - R, c_w + R]$ are divided into $L_r$ levels according to their relative squared distances to the center position. The position $(i, j) \in \Delta$ belongs to the $l$-th distance level if $R_{l-1} \le d_{i,j} < R_l$, where $d_{i,j} = (i - c_h)^2 + (j - c_w)^2$. From Eq. (2), we have $R_l = g^{l-1} R_1$. When the innermost radius $R_1$ is fixed, the LRF grows exponentially with the increase of $L_r$. The LRF is determined by $R$ which can be set arbitrarily. Given $R_{L_r} = R^2$ and $g$, we calculate $R_1 = max(2, R^2/g^{L_r-1})$. We use $R = \sqrt{R_{L_r}}$ as a hyperparameter instead of $R_1$, which is more flexible. Since we use the squared distance, we impose a minimum value of 2 to ensure that all 8-neighborhood pixels fall into the 1-st level.

All positions in the range of $\Delta$ are also uniformly divided into $L_\theta$ levels according to their relative directions from the center. The position $(i, j)$ belongs to the $m$-th level if $2\pi(m - 1)/L_\theta \le \theta_{i,j} < 2\pi m/L_\theta$, where $\theta_{i,j}$ is the counterclockwise angle from the vector $(0, 1)$ to the vector $(i - c_h, j - c_w)$. Combining the distance levels and the direction levels, the LRF is divided into $L_r \times L_\theta$ regions.

The LPSC kernel assigns a parameter to each region. All pixels of $X$ falling into the same region share the same parameter. For the region with the $l$-th distance level and $m$-th direction level, the assigned parameter is denoted by $w_{l,m}$. The areas of regions increase with $l$, the farther away from the center, the larger the area, the more pixels sharing parameters. Because the center position of the kernel is important and forms the basis of regions, we assign an additional separate parameter $w_{0,0}$ for the center pixel. A conventional kernel with a size of $(2R + 1) \times (2R + 1)$ has $(2R + 1)^2$ parameters, while a LPSC kernel only has $L_r \times L_\theta + 1$ parameters no matter how large $R$ is. When $R$ ranges from 2 to 9, a single conventional kernel has 25 to 361 parameters. In this range, it is sufficient to set $L_r$ to 2 or 3 and set $L_\theta$ to 6 or 8, so an LPSC kernel only has 13 to 25 parameters.

Let $N_{l,m}$ denote the number of pixels falling into the region $bin(l, m)$ with the $l$-th distance level and the $m$-th direction level. In faraway regions with large $l$, $N_{l,m}$, the impacts of pixels in them should be weakened. Therefore, we regularize the weight $w_{l,m}$ of each region by $N_{l,m}$: $w_{l,m}/N_{l,m}$. As a result, the LPSC kernel aggregates finer information from pixels nearing the center and is less sensitive to those of pixels farther away. Similar to conventional convolution, the LPSC kernel is slid along the input feature map $X$ with a pre-defined stride to perform convolution, as shown in Fig. 2(a).

When the kernel is located at a spatial location $(i, j)$, the output response is calculated as

$$(\boldsymbol{X} * \boldsymbol{W})(i, j) = \boldsymbol{W}(0, 0) \cdot \boldsymbol{X}(i, j) + \sum_{l=1}^{L_r} \sum_{n=1}^{L_\theta} \boldsymbol{W}(l, m) \cdot \left(\frac{1}{N_{l,m}} \sum_{u,v \in bin(l,m)} \boldsymbol{X}(u, v)\right) + b \quad (3)$$

For the LPSC kernel, the shape of its LRF is not necessarily a standard circle, but can be an oblique ellipse. As shown in Fig. 2(b), two additional hyper-parameters are introduced: the initial angle $\alpha$ and the eccentricity of the ellipse $e$. When dividing the regions, the distances are calculated according to the squared ellipse distance and the initial angle is added to the calculated directions. In this way, the LPSC kernel can better fit objects with different rotations and scales. In our experiments, we only evaluate the standard circular LRF by setting $\alpha = 0$ and $e = h/w = 1$.

## 3.2 LPSC via log-polar space pooling

Due to the special structure and parameter sharing, LPSC cannot be directly performed by popular deep learning frameworks. In this subsection, we show that LPSC can be readily implemented by conventional convolutions via log-polar space pooling to utilize efficient convolution modules.

Given the hyper-parameters $R$, $L_r$, $L_\theta$, and $g$ of the proposed LPSC, we can pre-compute a mask matrix $\boldsymbol{I}$ to indicate the region indexes of positions. The size of the mask $\boldsymbol{I}$ is $(2R + 1) \times (2R + 1)$. $1, \cdots, L_\theta \times L_r$ in $\boldsymbol{I}$ indicates the region index of the corresponding position. 0 indicates that the corresponding position does not fall into the LRF, since the region of the mask is the circumscribed rectangle of the LRF. The mask is slid through the input feature map $\boldsymbol{X}$ with the same stride of the LPSC convolution. As shown in Fig. 3(b), when the mask is located at a spatial location $(i, j)$, pixels of $\boldsymbol{X}$ in the range are divided into regions indicated by the mask. All pixels in the same region are encoded into a single pixel by mean pooling. We re-arrange the pooled pixels of different regions into a matrix of $2L_r \times L_\theta/2$ to preserve their relative spatial positions, as shown in Fig. 3(a). In this way, given $H' \times W'$ convolution locations ($H' = H$ and $W' = W$ if the stride is 1 with padding), the spatial size of the output map $\boldsymbol{X}_p$ after log-polar space pooling equals $2H'L_r \times W'L_\theta/2$.

We perform conventional convolution with $C'$ output channels on the output map $\boldsymbol{X}_p$ without padding. The size of the conventional convolution kernel is set to $(2L_r, L_\theta/2)$ and the stride is also $(2L_r, L_\theta/2)$. The output feature map $\boldsymbol{Y}_p$ has a size of $H' \times W' \times C'$. This is equivalent to performing the second term in Eq. (3). To model the first term, we use a separate $1 \times 1$ conventional convolution with the same $C'$ channels on the original $\boldsymbol{X}$. The stride is the same as the log-polar space pooling. The output feature map $\boldsymbol{Y}_c$ contains the convolution responses of the center pixels. We add this separate center pixel convolution output $\boldsymbol{Y}_c$ to the contextual convolution output $\boldsymbol{Y}_p$. $\boldsymbol{Y}_c + \boldsymbol{Y}_p$ serves as the output feature map of the proposed LPSC.

## 3.3 Incorporating LPSC into different CNNs

LPSC can be integrated into different CNN architectures. A straightforward way is to replace all conventional convolution kernels with LPSC kernels in a part of convolution layers. For plain CNN architectures such as AlexNet [2] and VGGNet [22], we simply perform this strategy in lower layers to increase the LRFs. However, some network architectures such as ResNet [23] are constituted of specifically designed blocks. In ResNet, either the bottleneck or the basicblock structure only contains $3 \times 3$ and $1 \times 1$ convolutions. Due to the difference in the local receptive field, the information captured by these small convolutions and LPSC may be different. In order to better incorporate these two types of information, we propose a cross convolution strategy as an alternative to replacing all convolutions in each layer of the block. Specifically, we set a ratio $p$. For each of several consecutive layers, we replace $p\%$ of all convolution kernels to LPSC kernels, while the remaining $(100 - p)\%$ of conventional kernels remain the same. In this way, each convolution kernel in the next layer, whether it is a conventional or an LPSC kernel, perceives the outputs generated by both the conventional and LPSC kernels of the previous layer. We denote this cross-convolution strategy by LPSC-CC. Details on how to incorporate LPSCs depend on the CNN architecture and will be presented in Section 4. Our code is available at `https://github.com/BingSu12/Log-Polar-Space-Convolution`.

### 3.4 Discussions

**Complexity.** For a $(2M+1) \times (2N+1) \times C$ kernel, conventional convolution involves $(2M+1) \times (2N+1) \times C$ multiplications and $(2M+1) \times (2N+1) \times C$ additions. LPSC with $L_r$ distance levels and $L_\theta$ direction levels only involves $2 * L_r \times L_\theta \times C$ multiplications, $(2M+1) \times (2N+1) \times C$ additions, and $(2M+1) \times (2N+1)$ lookups. The complexity of pre-computing the mask for lookup is $O(R^2)$, which only needs to be calculated once when initialing the layer. Typically, if $L_r = 2$, $L_\theta = 6$, LPSC only executes $24C$ multiplications for any size. However, even for a small $(2M+1) \times (2N+1) = 5 \times 5$ kernel, conventional convolution executes $25C$ multiplications; for a $9 \times 9$ kernel, multiplications increase to $81C$.

**Structural benefits.** With the special log-polar structure, the LPSC kernel naturally encodes the local spatial distribution of pixels w.r.t. the center and puts more attention to those adjacent pixels. Pixels with similar relative distances and directions share the same parameter, which not only reduces the number of parameters, but also makes the filter more robust and compact. Due to the logarithm effect, when located at different objects, small objects are relatively enlarged, while large objects are relatively reduced. Therefore, LPSC is less sensitive to the size of objects. Advantages of log-polar space pooling and extensions of LPSC to 1-D and 3-D data are discussed in the appendix.

**Relation with effective receptive field [11].** In [11], it is found that the ERF only occupies a fraction of the full theoretical receptive field. Specifically, the ERF size is $O(k\sqrt{n})$, where $k = 2R + 1$ is the kernel size and $n$ is the number of layers. Therefore, increasing the kernel size has a greater effect on expanding the ERF. It is also found that not all pixels in the LRF contribute equally, where the impacts of pixels near the center are much larger. The LPSC kernel follows this spirit to treat pixels near the center finely and increase the LRF exponentially.

**Drawbacks.** LPSC has two main drawbacks. (1) It introduces three additional hyper-parameters: $L_r$, $L_\theta$, and $g$. However, in practice, their selectable ranges are quite limited. Generally, to make the 8-neighborhoods of the center pixel have finer and non-redundant regional resolution, $L_r$ is set to 2 or 3, $L_\theta$ is set to 6 or 8, and $g$ is set to 2 or 3. (2) Its implementation via log-polar space pooling incurs large memory overhead. The space complexity of the upsampled feature map $\boldsymbol{X}_p$ is $O(H'W'L_rL_\theta C)$. For a single layer, the space complexity of LPSC is $O(H'W'L_rL_\theta C + L_rL_\theta CC' + H'W'C')$.

**Limitations.** Parameter sharing in LPSC aims to expand the local receptive field without increasing the number of parameters, but the cost is the loss of some fine-grained information. LPSC is more suitable for semantically sparse visual data that contains redundant information. As long as the data distribution conforms to the local correlation assumption, our LPSC can also be applied to irregularly sampled data, provided that the relative distances and angles between data points are defined. However, if the mask matrix to indicate the region indexes of positions cannot be pre-computed, the speed of LPSC will be very slow, because the region that each sampled data falls in should be calculated on-the-fly. LPSC may not be suitable for semantically dense data such as speech signals, text sequences, and amino acid sequences.

## 4 Experiments

### 4.1 Image classification experiments

For image classification, we evaluate the behaviors of LPSC integrated with different CNN architectures on three datasets: CIFAR-10, CIFAR-100 [52], and ImageNet [53]. We plug LPSC into three typical CNN architectures, including AlexNet [2], VGGNet-19 [22], and ResNet20 [23], by replacing a part of the conventional convolution layers. We use the Pytorch [54] implementation[2] of these architectures as our baseline. For the AlexNet, there are 5 convolution layers each followed by a ReLU activation layer. The sizes of the convolution kernels are $11 \times 11$, $5 \times 5$, $3 \times 3$, $3 \times 3$, and $3 \times 3$, respectively. For the VGG19 Net, there are sixteen convolution layers. The kernel size for all convolution layers is $3 \times 3$. For the ResNet-20, there are 9 basic blocks. Each block contains two $3 \times 3$ convolution layers. A $3 \times 3$ convolution layer is applied before all blocks. When the conventional convolutions in a layer or block are replaced by LPSCs, the number of kernels and the size of the output feature map remain the same as the original convolution layer.

---

[2]https://github.com/bearpaw/pytorch-classification

Table 1: Comparison of different convolution methods.

(a) Accuracy (%) with AlexNet and VGGNet-19

| Convolution | AlexNet | | | VGGNet-19 | | |
| --- | --- | --- | --- | --- | --- | --- |
| | Ori | Dilation | LPSC | Ori | Dilation | LPSC |
| # Params (M) | 2.47 | 2.34 | 2.31 | 20.04 | 20.08 | 20.08 |
| CIFAR-10 | 77.43 (0.25) | 75.42 (0.06) | **78.44** (0.12) | 93.54 (0.06) | 93.46 (0.14) | **93.92** (0.06) |
| CIFAR-100 | 43.98 (0.43) | 44.43 (0.10) | **47.43** (0.20) | 72.41 (0.17) | 73.03 (0.34) | **73.13** (0.12) |

(b) Accuracy (%) with ResNet-20

| Convolution | Ori | Dilation | LPSC | LPSC-CC |
| --- | --- | --- | --- | --- |
| # Params (M) | 0.27 | 0.27 | 0.27 | 0.27 |
| Acc. CIFAR-10 (%) | 91.66 (0.13) | 91.44 (0.10) | **91.81** (0.21) | **92.01** (0.08) |
| Acc. CIFAR-100 (%) | 67.56 (0.27) | 66.90 (0.25) | **67.63** (0.27) | **68.09** (0.27) |

Table 2: Results with ResNet-110 on CIFAR-100.

| Model | Inference Time | Acc |
| --- | --- | --- |
| Conv | 0.025 (0.0053) | 73.50 (0.24) |
| Dilation | 0.025 (0.0041) | 73.75 (0.48) |
| LPSC | 0.036 (0.0047) | 73.54 (0.38) |
| LPSC-CC | 0.135 (0.0035) | **74.13** (0.14) |

Table 3: Results with ResNet-18 on ImageNet.

| Model | #params(M) | Top-1 Acc (%) |
| --- | --- | --- |
| Conv | 11.69 | 69.86 (0.082) |
| LPSC | 11.69 | **69.96** (0.081) |
| LPSC-CC | 11.70 | 69.95 (0.079) |

To make a fair comparison, all experimental setup and details including the learning rate, batch size, number of filters per layer, hyper-parameters for the optimizer (e.g., $\gamma$, momentum, weight decay) remain exactly the same as in the baseline. We did not tune any of these setups for our LPSC. Therefore, the differences in performances only come from the changes in convolution layers. The numbers of parameters are computed on the CIFAR-10 dataset. Top-1 accuracy is used as the performance measure.

**Results on the CIFAR10 and CIFAR100 dataset.** We train the AlexNet, VGGNet-19, and ResNet-20 with conventional convolution, dilation convolution, and LPSC five times by using different random seeds for initialization, respectively, and compare the average accuracies and standard deviations. "Mean accuracy (standard deviation)" results are reported in Table 1. We use LPSC in the first two convolution layers for AlexNet, in the added first convolution before all blocks for VGGNet-19, and in the first convolution layer before all residual blocks for ResNet-20. Hyper-parameters of the LPSC kernels in different layers and networks are the same as the first three columns in Table A4(d) in the appendix, respectively. These choices are based on the ablation study as described in Appendix A.2 and A.3. For dilation convolution, we replace the conventional convolutions with dilation convolution in the same layers in the three architectures, respectively, where the kernel size and dilation rates are set so that the LRF and number of parameters are comparable with LPSC. Specifically, for AlexNet, the kernel size and dilation rate are set to 5 and 2 in the first convolution layer, respectively, and 4 and 2 in the second convolution layer, respectively. For VGGNet-19, the kernel size and dilation rate are set to 4 and 2 in the added first convolution layer before all blocks, respectively. For ResNet-20, the kernel size and dilation rate are set to 4 and 3 in the first convolution layer before all residual blocks, respectively. These choices are based on the evaluations in Table A4 of Appendix A.3. From Table 1, we observe that LPSC outperforms dilation convolutions with comparable LRF and parameters. The standard deviations for LPSC are limited, which shows that LPSC is not particularly sensitive to initializations. In some cases, the worst results also exceed those of the original networks with conventional convolutions and dilation convolutions by a margin.

We also evaluate the cross convolution strategy for ResNet-20. We apply LPSC-CC to the layer before all blocks and all $3 \times 3$ layers of the first block with a fixed $p$ of 50. From Table 1(b), we observe that the cross convolution strategy further improves the performances.

**Results with ResNet-110.** We train ResNet-110 with different convolutions on CIFAR-100 in Tab. 2. We follow the same setting for evaluating ResNet20, where $5 \times 5$ LPSC kernels ($L_r, L_\theta, g = 2, 6, 3$) are used to replace $3 \times 3$ convolutions in the first layer before all blocks in LPSC and in the first three layers with a fixed $p$ of 50 in LPSC-CC. For the deeper model, the advantage of LPSC is weakened, but LPSC-CC still improves ResNet110 significantly.

Table 4: Results on the VOC 2012 dataset. "-" means that the results are not reported. "*" indicates our reproduced results.

| Method | oAcc | mAcc | fAcc | mIoU |
|--------|------|------|------|------|
| DeepLabv3 | - | - | - | 0.701 |
| DeepLabv3+ | - | - | - | 0.711 |
| DeepLabv3+* | 0.9230 | 0.8332 | 0.8652 | 0.7144 |
| DeepLabv3+LPSC | **0.9273** | **0.8388** | **0.8714** | **0.7260** |

Table 5: Results on the DRIVE dataset [55]. "*" indicates our reproduced results.

| Method | Sen | Acc | AUC |
|--------|-----|-----|-----|
| HED [56] | 0.7364 | 0.9434 | 0.9723 |
| Azzopardi et al. [57] | 0.7655 | 0.9442 | 0.9614 |
| Zhao et al. [58] | 0.7420 | 0.9540 | 0.8620 |
| U-Net [59] | 0.7537 | 0.9531 | 0.9601 |
| Deep Vessel [60] | 0.7603 | 0.9523 | 0.9752 |
| CE-Net [61] | 0.8309 | 0.9545 | 0.9779 |
| CE-Net* [61] | 0.8266 (0.0106) | 0.9550 (0.0009) | 0.9782 (0.0008) |
| CE-Net-LPSC-1 | 0.8300 (0.0079) | **0.9552** (0.0011) | 0.9782 (0.0008) |
| CE-Net-LPSC-2 | **0.8312** (0.0075) | 0.9548 (0.0011) | **0.9784** (0.0007) |

**Comparison of FLOPs.** Comparisons of the average runtime per batch for using different convolutions in ResNet110 are shown in Tab 2. LPSC runs slower than conventional convolution, but this is because we use of-the-shell conventional convolution modules in Pytorch to implement LPSC, which are highly optimized and very efficient for conventional convolution. LPSC can be greatly accelerated if it can be directly implemented with CUDA or by directly adapting the underlying code of convolutions in the integrated framework. On CIFAR10 with AlexNet, the FLOPs (recorded by the fvcore toolbox[3]) of conventional convolution, dilated convolution, and LPSC are 14.95M, 24.71M, and 11.42M, respectively. LPSC has much lower FLOPs than other convolution methods.

**Results on the ImageNet dataset.** ImageNet [53] contains 1.28 million training images and 50k validation images from 1000 classes. We again use the Pytorch implementation[4] of ResNet-18 as the baseline. For LPSC, we replace conventional convolution with LPSC in the first convolution layer before all blocks of ResNet-18, where the size $2R + 1$, $L_r$, $L_\theta$, and $g$ for LPSC kernels are 9, 3, 8, and 2, respectively. For LPSC-CC, in addition to reduce $p$ from 100 to 25 in the first layer, we also replace a quarter of $3 \times 3$ kernels with LPSC kernels in the first residual block (i.e., $p = 25$), where the size $2R + 1$, $L_r$, $L_\theta$, and $g$ for LPSC kernels in the block are 5, 2, 6, and 3, respectively. The setting of these hyper-parameters for LPSC follows the suggestions in the ablation study in Appendix A.2. Due to the limitation of computing resources, we reduced the batch size and learning rate by 4 times. Other hyper-parameters remain the same. We compare the mean top-1 accuracy and the standard deviation of the last ten epochs in Tab. 3. Both LPSC and LPSC-CC slightly improve the top-1 accuracy and the standard deviation of ResNet-18.

## 4.2   Semantic segmentation experiments

LPSC can also be applied to other vision tasks. On the PASCAL VOC 2012 dataset [62, 63] for general image semantic segmentation, we adopt the Pytorch implementation[5] of DeepLabv3+ [64] with the MobileNet [65] backbone as the baseline. The training set is augmented by extra annotations provided in [66]. Overall accuracy (oAcc), mean accuracy (mAcc), freqw accuracy (fAcc), and mean IoU (mIoU) on the validation set are evaluated. In DeepLabv3+, the atrous spatial pyramid pooling (ASPP) module probes multi-scale features by applying atrous/dilated convolutions with three different rates. For DeepLabv3+LPSC, we replace the dilated convolution with the largest rate by LPSC in ASPP. The kernel size, $L_r$, $L_\theta$, and $g$ of LPSC are set to 9, 2, 8, 2, respectively. Comparisons with the reported and reproduced results are shown in Tab. 4. LPSC improves DeepLabv3+ by a margin of 1.1% on mIoU. All hyper-parameters and setups such as the learning rate, batch size, etc, remain the same, so the performance gains are only attributed to the proposed LPSC.

---

[3]https://github.com/facebookresearch/fvcore
[4]https://github.com/bearpaw/pytorch-classification
[5]https://github.com/VainF/DeepLabV3Plus-Pytorch

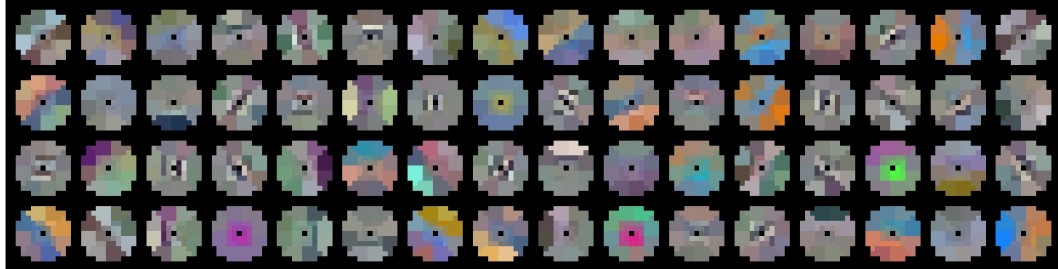

Figure 4: Visualization of the learned circular LPSC kernels without center convolution in the first convolution layer of Alexnet on the CIFAR-10 dataset.

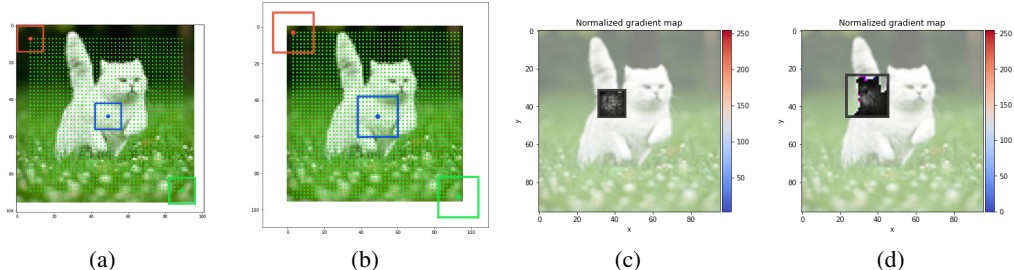

| (a) | (b) | (c) | (d) |

Figure 5: Estimated RFs with (a) conventional convolution and (b) LPSC. The normalized gradient map with (c) conventional convolution and (d) LPSC at a sampled location.

On the DRIVE dataset [55] for retinal vessel detection, we adopt CE-Net [61] as the baseline. Sensitivity (Sen), accuracy (Acc), and AUC are evaluated on the test set. The dense atrous convolution (DAC) block of CE-Net uses four cascade branches with increasing numbers of dilated convolutions. For CE-Net-LPSC-1, we replace the dilated convolutions with rates of 3 and 5 by LPSCs with sizes of 5 and 11 in DAC, respectively, so that LPSCs have the same LRFs with dilated convolutions. $L_r$, $L_\theta$, and $g$ of LPSCs are set to 2, 6, 3, respectively. For CE-Net-LPSC-2, we increase the kernel sizes of LPSCs to 9 and 15, respectively, to further increase LRFs. We accordingly use more parameters by setting $L_r$, $L_\theta$, and $g$ to 3, 8, 1.5, respectively. Other hyper-parameters remain the same[6]. We run our models ten times and report the average performances. Comparisons with the reported results are shown in Tab. 5. Our LPSC achieves good generalization performances on medical image segmentation with limited training samples.

## 4.3 Visualization

**Visualization of the learned LPSC kernels.** In Fig. 4, we visualize the learned LPSC kernels in the first convolution layer of AlexNet on the CIFAR-10 dataset. The $11 \times 11$ LPSC kernels have 3 distance levels and 8 direction levels. In LPSC kernels, the closer to the center, the higher the regional resolution; the more outward, the larger the range for parameter sharing. We observe that the learned LPSC kernels capture some special local structures and contextual configuration. In some kernels, the weights for adjacent regions are continuous; some kernels are sensitive to specific directions, edges, colors, or local changes; in some other kernels, specific combinations of regions are highlighted. More visualizations are shown in Appendix A.4.

**Comparison of effective receptive field (ERF):** Fig. 5(a) and (b) show the estimated RFs of SimpleVGGNet on the default example using conventional convolutions and LPSCs in the first two layers by the gradient-based RF estimation[7], respectively. LPSC enlarges the estimated RFs from $14 \times 14$ to $22 \times 22$. The normalized gradient maps w.r.t. a position of the output for estimating the RF using conventional convolutions and LPSCs are shown in Fig. 5(c) and (d). With LPSC, gradients can be back-propagated to more pixels of the input image.

---

[6]https://github.com/Guzaiwang/CE-Net
[7]https://github.com/fornaxai/receptivefield

# 5 Conclusion

In this paper, we have presented LPSC that naturally encodes the local contextual structures. LPSC distinguishes regions with different distance levels and direction levels, reduces the resolution of remote regions, and reduces the number of parameters by weight sharing for pixels in the same region. The LRF of LPSC increases exponentially with the number of distance levels. We impose a regularization on the parameters and implement LPSC with conventional convolutions by log-polar space pooling and separable center pixel convolution. We analyze the interests and drawbacks of LPSC from different aspects. We empirically show the effectiveness of the proposed LPSC on five datasets for classification and segmentation tasks.

## Acknowledgments

The authors would like to thank the anonymous reviewers for their valuable comments. This work was supported in part by the National Natural Science Foundation of China No. 61976206 and No. 61832017, Beijing Outstanding Young Scientist Program NO. BJJWZYJH012019100020098, Beijing Academy of Artificial Intelligence (BAAI), the Fundamental Research Funds for the Central Universities, the Research Funds of Renmin University of China 21XNLG05, and Public Computing Cloud, Renmin University of China. This work was also supported in part by Intelligent Social Governance Platform, Major Innovation & Planning Interdisciplinary Platform for the "Double-First Class" Initiative, Renmin University of China, and Public Policy and Decision-making Research Lab of Renmin University of China.

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
