# Log-Polar Space Convolution Layers: Appendix

**Bing Su, Ji-Rong Wen**[*]
Beijing Key Laboratory of Big Data Management and Analysis Methods
Gaoling School of Artificial Intelligence, Renmin University of China
Beijing 100872, China
`subingats@gmail.com; jrwen@ruc.edu.cn`

## A  Appendix

### A.1  Statistics of correlations between different regions and the center pixel

We calculate the correlations between image pixels in different log-polar regions and the center pixels on the training set of CIFAR-100. Specifically, for each pixel in each image, we divide its $11 \times 11$ neighboring area into different regions by LPSC with 3 distance levels, 8 direction levels, and a growth rate of 2. The center pixels of all areas form the center set. The pixels at the same position of all areas also form a pixel set. For each position, we calculate the correlation score between the corresponding pixel set and the center set. The correlation scores of positions in the same region of all training images are averaged to obtain the correlation score between the region and the center pixel. In this way, we obtain the correlation scores from all $3 \times 8$ regions to the center pixel, as shown in Table A1. In further regions, the average correlation of pixels is lower, and the average correlations within different regions of the same distance level are comparable. The farther the region is from the center point, the lower the correlation, and the decay is rapid (approximately exponentially decreasing). These statistics verify the motivation of the log-polar space convolution.

### A.2  Ablation study

**Influence of hyper-parameters.** The proposed LPSC kernel has four hyper-parameters: the kernel size $2R + 1$, the number of distance levels $L_r$, the number of direction levels $L_\theta$, and the growth rate $g$. For small kernels, since we set a minimum value 2 for the smallest distance level, the largest $L_r$ can be determined by $R^2$ and $g$. If $L_r$ is too large, no pixels will fall into regions of large distance levels. $L_\theta$ can be set to 6 or 8, since in most cases there are only about 8 pixels in the smallest circle in Fig. 1(b) in the main text. We evaluate the influences of $L_r$, $L_\theta$, and $g$ by replacing the large $11 \times 11$ convolution kernels with the LPSC kernels in the first layer of AlexNet in Tab. A2(a) and by applying LPSC kernels with different sizes in the first convolution layer before all blocks of VGGNet and ResNet in Tab. A2(c) and (d), respectively. We run all models for only one time. Increasing the values of $L_r$ and $L_\theta$ results in finer regions and improves the performances, but the number of parameters also increases.

VGGNet and ResNet use small $3 \times 3$ kernels. To make the number of parameters comparable, we fix $(L_r, L_\theta)$ to $(2, 6)$ and evaluate the influence of the kernel size $2R + 1$ in Tab. A2(c)(d). When $2R + 1$ is too small, the LRF is limited. When $2R + 1$ is too large, the regions with large distance levels may be coarse, i.e., a large amount of positions share the same weight, which may decrease the resolutions of parameters. Overall, for large kernels ($11 \times 11$), we can set $(L_r, L_\theta, g)$ to $(3, 8, 2)$. For small kernels ($5 \times 5$), we may fix $(L_r, L_\theta, g)$ to $(2, 6, 3)^2$.

---

[*]Corresponding author: Ji-Rong Wen.

[2]In this case, since we set a lower bound of the first level distance, $g = 2$ and 3 will result in the same LPSC kernel.

Table A1: Correlation scores between different regions and the center pixel.

| Direction level | 4 | 3 | 2 | 1 | 5 | 6 | 7 | 8 |
|---|---|---|---|---|---|---|---|---|
| Distance level 1 | 0.6795 | 0.8268 | 0.6794 | 0.9162 | 0.8323 | 0.6799 | 0.8277 | 0.6800 |
| Distance level 2 | 0.4663 | 0.5844 | 0.4620 | 0.5967 | 0.5967 | 0.4620 | 0.5852 | 0.4665 |
| Distance level 3 | 0.2113 | 0.2779 | 0.1992 | 0.2983 | 0.2976 | 0.1968 | 0.2764 | 0.2097 |

Table A2: Ablation study based on (a)(b) AlexNet (c) VGGNet-19 and (d) ResNet-20.

(a) $11 \times 11$ LPSC kernels with different hyper-parameters are used in the first convolution layer of AlexNet.

| $(L_r, L_\theta, g)$ | (3,8,2) | (3,6,2) | (2,8,3) | (2,8,2) |
|---|---|---|---|---|
| # Params (M) | 2.45 | 2.45 | 2.45 | 2.45 |
| Acc. CIFAR-10 (%) | 77.27 | 76.83 | 76.23 | 75.95 |
| Acc. CIFAR-100 (%) | 46.35 | 45.86 | 45.07 | 44.61 |

(b) Effects of using LPSCs in different convolution layers of AlexNet. Kernel size remains the same as in the corresponding layer. $(L_r, L_\theta, g)$ is fixed to 3, 8, 2 and 2, 6, 3 when applied to the 1st and 2nd layer, respectively.

| LPSC layer | 1 | 2 | 1 + 2 |
|---|---|---|---|
| # Params (M) | 2.45 | 2.33 | 2.31 |
| Acc. CIFAR-10 (%) | 77.27 | **78.31** | 78.28 |
| Acc. CIFAR-100 (%) | 46.35 | 44.81 | **47.31** |

(c) LPSCs with different sizes and hyper-parameters are used in an additionally added convolution layer before all blocks in VGGNet-19.

| $2R + 1$ | 5 | 9 | 13 | 17 |
|---|---|---|---|---|
| $(L_r, L_\theta, g)$ | (2,6,3) | (2,6,3) | (2,6,3) | (3,8,4) |
| # Params (M) | 20.08 | 20.08 | 20.08 | 20.08 |
| Acc. CIFAR-10(%) | 93.66 | **94.01** | 93.86 | 93.73 |
| Acc. CIFAR-100(%) | 72.95 | 73.13 | 73.08 | **73.37** |

(d) Effects of using LPSCs in different layers and blocks of ResNet-20. The first two rows denote the sizes and hyper-parameters of LPSCs in the first convolution layer before all blocks. The third row denotes the sizes of LPSCs with fixed hyper-parameters (2,6,2) in all blocks. "-" means that no LPSCs are used in blocks. "B" means that two successive $3 \times 3$ convolution layers are replaced with a single LPSC layer in the BasicBlock.

| $2R + 1$ | 5 | 9 | 13 | 5 | 13 |
|---|---|---|---|---|---|
| $(L_r, L_\theta, g)$ | (2,6,2) | (2,6,3) | (2,6,3) | (2,6,2) | (2,6,2) |
| $2R + 1$ for Blocks | - | - | - | 5 | 9(B) |
| # Params (M) | 0.27 | 0.27 | 0.27 | 0.39 | 0.18 |
| Acc. CIFAR-10 (%) | 91.55 | 91.67 | **92.11** | 91.82 | 89.78 |
| Acc. CIFAR-100 (%) | 67.23 | 67.19 | 66.97 | **67.98** | 65.10 |

**Influence of the plugged layer.** Tab. A2(b) and (d) show the results of using LPSC in different layers or blocks for AlexNet and ResNet-20, respectively. It seems that performing LPSC in low layers is more beneficial. This may be because pixels in high layers have merged information from large LRFs so that even adjacent pixels may have different influences on the center pixel and the weights for different positions are not suitable for sharing. Applying LPSC in low layers is conducive to increase the LRFs, filter redundant details, and back-propagate the gradients to more bottom pixels. In the last column of Tab. A2(d), the two successive convolution layers in each block are replaced with a single LPSC layer, so the number of convolution layers is reduced by half and the number of parameters is reduced by one third, but the performances are only reduced by 2% using half the layers with LPSC in ResNet.

Table A3: Effects of the weight regularization and center pixel convolution based on AlexNet.

| Method | Sum | Max | No CenterConv | Mean |
|---|---|---|---|---|
| Acc. CIFAR-10(%) | 21.61 | 76.65 | **78.51** | 78.28 |
| Acc. CIFAR-100(%) | 5.53 | 44.63 | 47.13 | **47.31** |

**Effects of weight regularization.** In Tab. A3, we evaluate the effects of the weight regularization in Eq. (3) in the main text based on AlexNet. "Sum" shows the results by using sum pooling instead of mean pooling in log-polar space pooling in the first two LPSC layers. This is equivalent to remove the regularization and the performances are severely degraded. This is because far regions are exponentially larger than nearer regions. If positions in all regions are treated equally, even the weight for a far region is not too large, the accumulation of less relevant distant pixels will still produce an overwhelming response. We also try max pooling. It also performs worse than mean pooling. Due to the large LRF, regions with large distance levels for many adjacent center locations will have large overlaps. Some large responses may dominant repeatedly in many regions for different center locations, which suppress other useful local information.

**Effects of center pixel convolution.** In the fourth column of Tab. A3, we remove the center pixel convolution, i.e., the first term in Eq. (3) in the main text. Center pixel convolution enlarges the importance of the center pixel. Contextual information itself may be sufficient for classification when there are few classes. For more complex tasks with more classes, center pixel convolution may provide complementary information.

**Running times.** On the CIFAR10 dataset, the training time for one epoch and the testing time for AlexNet are 3.4808 and 0.9083, respectively; after replacing conventional convolutions with LPSCs in the first two layers, the training time for one epoch and the testing time are 23.3981 and 3.7476, respectively. In our implementation, LPSC runs much slower than conventional convolution, but this is because we use of-the-shell conventional convolution modules to implement LPSC. To this end, we must first apply log-polar space pooling with the fold and unfold operations in Pytorch, which consume much time and space complexity. LPSC can be greatly accelerated if it is directly implemented with CUDA or by directly adapting the underlying code of convolutions in the integrated framework.

## A.3  Comparison with other convolution methods

Dilated convolution [1] can also exponentially increase the LRF without increasing the number of parameters. Circle convolution [2] employs circle kernels and uniformly samples in the polar space. Deformable convolution [3] adaptively adjusts the LRF by using additional parameters to infer the offsets for each position. We use the Pytorch implementation of deformable convolution[3] in our experiments. Different from LPSC where regions are divided in the log-polar space, we can also averagely divide the kernel into different square regions, e.g., a $9 \times 9$ kernel can be divided into $3 \times 3$ regions with a size of $3 \times 3$. All positions in the same square region share the same parameter. We denote this alternative convolution method by *square convolution*, which also increases LRF with fewer parameters. Similarly, they can also be used to replace conventional convolution at different layers in different architectures.

Evaluations of these compared convolutions with different hyper-parameters are presented in Tab. A4. For dilation convolution, square convolution, and LPSC, the second and third rows show the hyper-parameters of the corresponding kernels in the first convolution layer and in other layers or blocks, respectively. We run all models for only one time. "Size" indicates the kernel size. For VGGNet and ResNet, "1+" indicates the first convolution layer before all blocks. The fourth row indicates the indexes of layers or blocks in which conventional convolution is replaced by the corresponding convolution. The third column blocks show the results of dilated convolution with different hyper-parameters in different layers. The hyper-parameters, including the kernel size and the dilation rate, are set to keep the total number of parameters and the LRF comparable to conventional convolution and LPSC.

Based on these evaluations, in Tab. A5, we compare LPSC with conventional convolution, dilation convolution, deformable convolution, circle convolution, and square convolution in the three architec-

---

[3]https://github.com/oeway/pytorch-deform-conv

Table A4: Performance of other convolution methods based on different architectures.

(a) AlexNet

| Conv Type | Circle | Dilation (size, dilation rate) | | | | | Deformable | | | | Square (size, pool size) | | |
|---|---|---|---|---|---|---|---|---|---|---|---|---|---|
| hy.-para. L-1 | 7 | 5,3 | 7,2 | - | - | 7,2 | | | | | 11,5 | - | 11,5 |
| hy.-para. oth. | 5 | - | - | 3,2 | 3,3 | 3,3 | | | | | - | 9,3 | 9,3 |
| Layer | 1,2 | 1 | 1 | 2 | 2 | 1,2 | 1 | 2 | 1,2 | all | 1 | 2 | 1,2 |
| #params(M) | 2.46 | 2.45 | 2.46 | 2.28 | 2.28 | 2.26 | 2.47 | 2.55 | 3.21 | 7.04 | 2.45 | 2.28 | 2.26 |
| CIFAR10 | 10 | 73.43 | 75.7 | 74.94 | 78.11 | 75.95 | 75.98 | 55.34 | 32.96 | 55.22 | 76.10 | 75.17 | 73.26 |
| CIFAR100 | 1 | 44.86 | 45.98 | 41.08 | 44.21 | 44.26 | 41.96 | 30.30 | 30.36 | 30.82 | 44.50 | 42.43 | 41.89 |

(b) VGGNet-19

| Conv Type | Circle | Dilation (size, dilation rate) | | | | Deformable | | | | Square (size, pool size) | | |
|---|---|---|---|---|---|---|---|---|---|---|---|---|
| hy.-para. L-1+ | 5 | - | - | - | - | | | | | 13,4 | 9,3 | 9,3 |
| hy.-para. oth. | - | 3,2 | 3,2 | 3,2 | 3,2 | | | | | - | - | 5,3 |
| Block | 1+ | 1 | 2 | 1,2 | 1,2,3 | 1+ | 1+,1 | 1+,1,2 | 1+,all | 1+ | 1+ | 1+,1 |
| #params(M) | 20.08 | 20.04 | 20.04 | 20.04 | 20.04 | 20.04 | 20.11 | 20.48 | 58.53 | 20.04 | 20.04 | 20.04 |
| CIFAR10 | 93.85 | 91.53 | 91.97 | 89.61 | 89.56 | 92.53 | 70.37 | 90.64 | 90.02 | 87.42 | 90.01 | 89.4 |
| CIFAR100 | 72.76 | 68.46 | 69.30 | 63.75 | 63.83 | 69.32 | 37.50 | 64.37 | 61.26 | 62.51 | 66.64 | 65.79 |

(c) ResNet-20

| Conv Type | Circle | Dilation (size, dilation rate) | | | | Deformable | | Square (size, pool size) | | | |
|---|---|---|---|---|---|---|---|---|---|---|---|
| hy.-para. L-1+ | 5 | 3,2 | 3,3 | 5,2 | 3,2 | | | 13,4 | 9,3 | 9,3 | 9,3 |
| hy.-para. oth. | - | - | - | - | 3,2 | | | - | - | 5,3 | 5,3 |
| Block | 1+ | 1+ | 1+ | 1+ | 1+,1,2,3 | 1+ | 1,2,3 | 1+ | 1+ | 1+,1 | 1+,1,2 |
| #params(M) | 0.27 | 0.27 | 0.27 | 0.27 | 0.27 | 0.27 | 0.78 | 0.27 | 0.27 | 0.27 | 0.28 |
| CIFAR-10 | 91.76 | 91.39 | 91.77 | 91.33 | 86.09 | 90.27 | 46.57 | 88.09 | 89.84 | 88.43 | 87.19 |
| CIFAR-100 | 67.58 | 66.95 | 67.58 | 67.08 | 59.26 | 65.44 | 13.45 | 61.43 | 63.54 | 62.05 | 60.37 |

(d) Results of LPSC with different architectures

| Architecture | AlexNet | VGGNet | ResNet $(size, L_r, L_\theta, g)$ | |
|---|---|---|---|---|
| hy.-para. L-1+ | 11,3,8,2 | 9,2,6,3 | 13,2,6,3 | 5,2,6,2 |
| hy.-para. oth. | 9,2,6,3 | - | - | 5,2,6,2 |
| Layer/Block | 1,2 | 1+ | 1+ | 1+,all |
| #params(M) | 2.31 | 20.08 | 0.27 | 0.39 |
| CIFAR-10 | **78.28** | **94.01** | **92.11** | 91.82 |
| CIFAR-100 | **47.31** | **73.13** | 66.97 | **67.98** |

tures, respectively. In AlexNet and VGGNet, LPSC outperforms dilated convolution with different hyper-parameters significantly. This shows the effectiveness of spatial structure and parameter sharing in LSPC. Applying dilated convolution in higher layers also leads to worse performances, which further verifies our analysis in Sec. A.2, but LPSC outperforms dilated convolution. When only used in the first layer of ResNet, sometimes dilated convolution achieves slightly better results than LPSC. The reason may be that ResNet can stack deeper layers with residual connections and hence achieve large LRF using multi-layer small regular kernels, which eases the need for large LRF in lower layers. Moreover, the non-uniform distribution of parameters in LPSC may cause information dispersion and over-smoothing, therefore it is more difficult to model residuals.

Deformable convolution introduces additional parameters. When applied to all blocks of VGGNet or ResNet, the parameters are more than doubled. Deformable convolution causes performance degradation of different architectures. LPSC also outperforms the circle convolution and the average square convolution. This indicates that the spatial structure designed in log-polar space can better capture the contextual information.

Table A5: Comparison of different convolution methods.

(a) Accuracy (%) with AlexNet

| Convolution | Ori | Dilation | Deformable | Circle | Square | LPSC |
|---|---|---|---|---|---|---|
| # Params (M) | 2.47 | 2.34 | 2.47 | 2.46 | 2.45 | 2.31 |
| CIFAR-10 | 77.43 (0.25) | 75.42 (0.06) | 75.98 | 10 | 76.10 | **78.44** (0.12) |
| CIFAR-100 | 43.98 (0.43) | 44.43 (0.10) | 41.96 | 1 | 44.50 | **47.43** (0.20) |

(b) Accuracy (%) with VGGNet-19

| Convolution | Ori | Dilation | Deformable | Circle | Square | LPSC |
|---|---|---|---|---|---|---|
| # Params (M) | 20.04 | 20.08 | 20.04 | 20.08 | 20.04 | 20.08 |
| CIFAR-10 | 93.54 (0.06) | 93.46 (0.14) | 92.53 | 93.85 (0.08) | 90.01 | **93.92** (0.06) |
| CIFAR-100 | 72.41 (0.17) | 73.03 (0.34) | 69.32 | 72.76 (0.28) | 66.64 | **73.13** (0.12) |

(c) Accuracy (%) with ResNet-20

| Convolution | Ori | Dilation | Def. | Circle | Square | LPSC | LPSC-CC |
|---|---|---|---|---|---|---|---|
| # Params (M) | 0.27 | 0.27 | 0.27 | 0.27 | 0.27 | 0.27 | 0.27 |
| CIFAR-10 | 91.66 (0.13) | 91.44 (0.10) | 90.27 | 91.76 (0.14) | 89.84 | **91.81** (0.21) | **92.01** (0.08) |
| CIFAR-100 | 67.56 (0.27) | 66.90 (0.25) | 65.44 | 67.58 (0.28) | 63.54 | **67.63** (0.27) | **68.09** (0.27) |

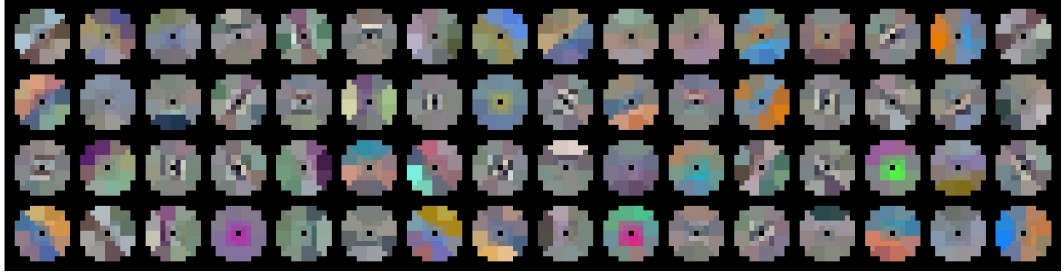

(a)

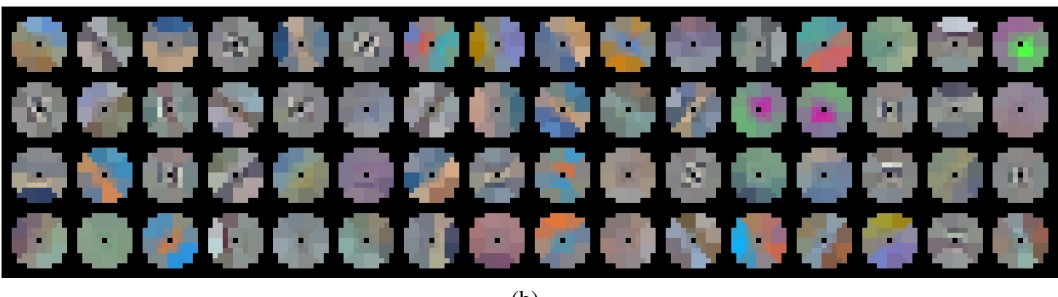

(b)

Figure A1: Visualization of the learned circular LPSC kernels without center convolution in the first convolution layer of Alexnet on (a) the CIFAR-10 dataset and (b) the CIFAR-100 dataset.

## A.4   Visualization

**Visualization of the learned LPSC kernels.** In Fig. A1, we visualize the learned LPSC kernels in the first convolution layer of AlexNet on the CIFAR-10 and CIFAR-100 datasets. The LPSC kernels have a size of $11 \times 11$, 3 distance levels, 8 direction levels, and a growth factor of 2. Since LPSC kernels in the first layer have three channels, we normalize the values of kernels into the range of [0, 255] and view each position of the kernel as an RGB pixel. Different from conventional convolution kernels, in LPSC kernels, the closer to the center, the higher the regional resolution; the more outward, the larger the range for parameter sharing. We observe that the learned LPSC kernels capture some special local structures and contextual configuration. In some kernels, the weights for adjacent regions are continuous; some kernels are sensitive to simple directions and edges, some others are sensitive to complex boundaries and color combinations, and in some other kernels, specific combinations of regions are highlighted. To fully utilize the space circumscribed by the LRF of the kernel, we fill four

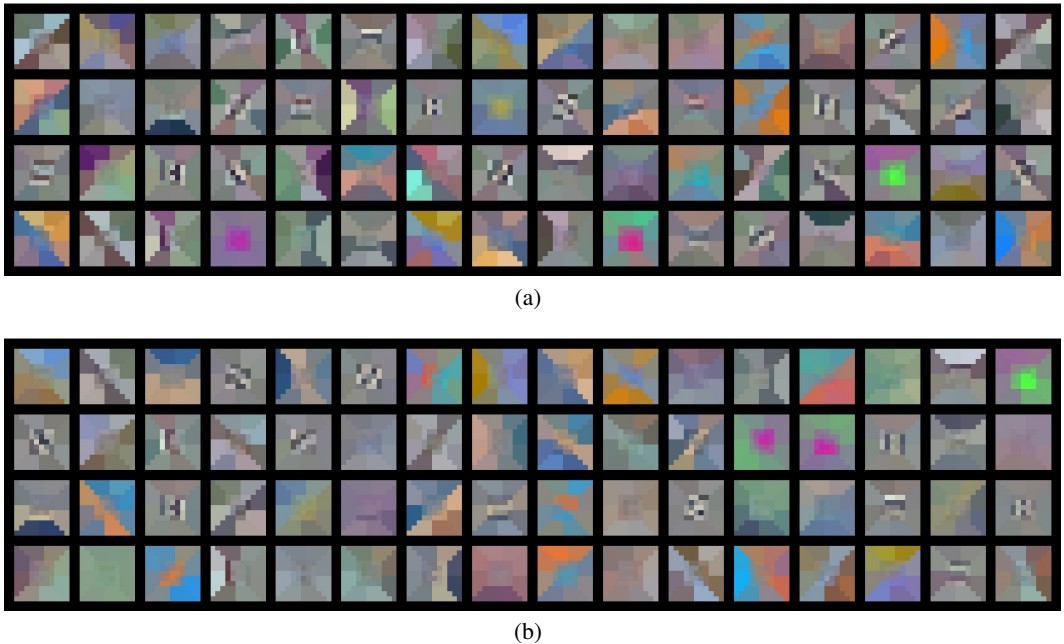

(a)

(b)

Figure A2: Visualization of the filled $11 \times 11$ LPSC kernels without center convolution in the first convolution layer of Alexnet on (a) the CIFAR-10 dataset and (b) the CIFAR-100 dataset.

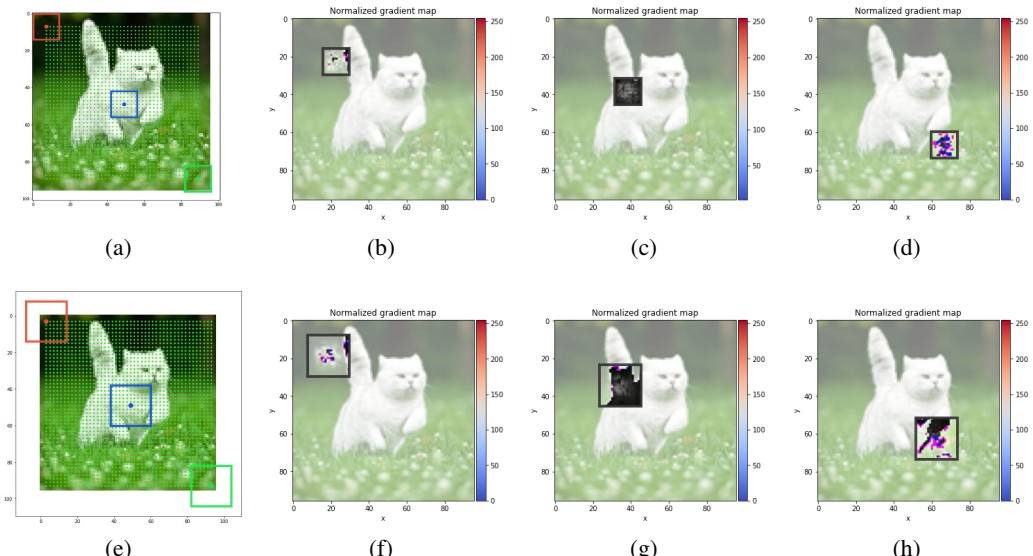

Figure A3: Estimated RFs with (a) conventional convolution and (e) LPSC. The normalized gradient map with (b-d) conventional convolution and (f-h) LPSC at different locations.

corners (positions that do not fall into the LRF) with the weights of corresponding nearest regions in all experiments, respectively, as shown in Fig. A2.

**Comparison of effective receptive field (ERF):** Fig. A3(a) and (e) show the estimated RFs of SimpleVGGNet on the default example using conventional convolutions and LPSCs in the first two layers by the gradient-based RF estimation[4], respectively. LPSC enlarges the estimated RFs from $14 \times 14$ to $22 \times 22$. The normalized gradient maps w.r.t. different positions of the output for estimating

---

[4]https://github.com/fornaxai/receptivefield

the RFs using conventional convolutions and LPSCs are shown in Fig. A3(b-d) and Fig. A3(f-h), respectively. With LPSC, gradients can be back-propagated to more pixels of the input image.

## A.5 Additional discussions

**Advantages of log-polar space pooling.** In log-polar space pooling, we use mean pooling to achieve the regularization for regions. In this way, LPSC does not increase the effective capacity of the model since only mean pooling within each region is performed w.r.t. the average operation in the innermost parentheses of Eq.(3) in the main text and no learnable parameters are involved in log-polar space pooling. If we learn parameters rather than performing pooling in the log-polar space pooling step, much more parameters will be introduced. In this step, other pooling methods are also allowed. For example, if sum pooling is used, it is equivalent to remove the regularization. If max pooling is used, LPSC can be viewed as a special dilated convolution with irregular and data-driven holes.

LPSC cannot be viewed as a special case of separable convolution [4], since after log-polar space pooling we are not performing $1 \times 1$ convolutions. On the contrary, LPSC can be combined with separable convolution, i.e., conventional convolutions can be replaced by separable convolutions after log-polar space pooling. When we use LPSC to replace the conventional convolution, after log-polar pooling, kernels with a comparable amount of parameters as the original convolution layer are used. Therefore, log-polar pooling can be approximately viewed as being additionally inserted into the original architecture to increase the receptive field and non-linearly aggregate large-range contextual information.

**Relation with advanced convolution methods.** For LPSC, parameters distribute in the log-polar space. Most advanced convolution methods discussed in Section 2 of the main text do not change the regular sampling grid and focus on different aspects with LPSC. Therefore, these advances may be propagated to LPSC as well. For example, group and separable convolution [5, 4] separates the channel dimension into groups and hence can be combined with the LPSC kernel in spatial dimensions. Graph convolution can also take advantage of LPSC as long as the relative distance and direction between nodes are defined. The solution in [6] to the coordinate transform problem can also be directly applied to our LPSC by putting Cartesian coordinate into pixel.

**Extending LPSC to tackle 1-D and 3-D data.** LPSC kernel can directly degenerate to 1-D kernel, where there are only two directions (forward and backward) with respect to the center point and the kernel is divided into different distance levels along each direction, e.g., regions of the kernel in Fig. 1(b) along the vertical line are extracted to form a $1 \times 7$ 1D LPSC kernel $[w_{32}, w_{32}, w_{12}, w_{00}, w_{15}, w_{25}, w_{35}]$. There are two possible ways to extend LPSC to 3-D kernels. (1) Cylindrical kernel: For a $h \times w \times t$ kernel, all $t$ 2-D sub-kernels with a size of $h \times w$ are divided into different regions as in Fig. 1(b), and regions at the same location in all $t$ sub-kernels share the same weight. (2) Spherical kernel: distance levels can be naturally extended to 3 dimensions (similar to a small ball within a big ball); an additional direction dimension (such as a plane angle) is introduced to divide the direction (similar to cutting a watermelon). We will explore such extensions in our future work.

**Possible benefits of ellipsoid LPSC kernels.** In natural images, different objects or local parts have different shapes and orientations, so it may be suitable to use ellipsoid LPSC kernels to fit objects or parts with different orientations and sizes. For instance, in image understanding, we can fuse multiple ellipsoid LPSC kernels with different sizes (local receptive fields) and orientations in a multi-channel parallel manner. Moreover, a given ellipsoid LPSC kernel can be scaled without changing the parameters. If the orientations and size scaling of LPSC kernels can be learned while learning the convolution parameters, rotation and scale invariant features can be extracted from objects or local parts, so that the learned representations are more robust.