# OpenReview forum: "Log-Polar Space Convolution Layers"
_NeurIPS.cc/2022/Conference — NeurIPS 2022 Accept_

### Official Review · Reviewer_AZ8Z · 2022-06-29

**Rating:** 7
**Confidence:** 3
**Soundness:** 3 good
**Presentation:** 3 good
**Contribution:** 4 excellent

**Summary:**


In conventional CNNs, large effective receptive fields (ERF) are generally achieved through a cascading of multiple layers of convolutions and pooling operations. The authors note that this requirement may have an impact on the model performance; it may cause vanishing gradients and moreover may not actually have the desired effect of an increased ERF. The effective receptive field theory states that a more effective way of increasing ERF is increasing the kernel size. To this end, the authors explore a parameter efficient way of increasing through defining their kernels in log-polar space. The authors carefully describe and motivate their method and its relation to conventional convolutions, and experimentally show the validity of their method on a range of experiments. Furthermore, the authors show a compute-efficient implementation using conventional convolution operations.

**Questions:**

1. Do you have an intuition for when you would want to use ellipsoid LPSCs? Are there any specific use-cases that come to mind?
2. Your method actually gives a definition for kernel values over the entire continuous input space. Does this mean your method could be readily applied to irregularly sampled data? Are there any specifics of your method that would inhibit you from doing this? Have you attempted this?
3. The authors motivate their log-polar space kernel definition by saying that in natural images often correlation between local pixels is higher than correlation between more distant pixels and therefore we can assign the same parameters to larger patches of pixels further away. I don’t think this argument necessarily makes sense; indeed natural images are locally correlated, but I believe this only says that local pixels should be assigned larger weights values, not that larger patches of pixels further away should be assigned identical weight values. Indeed, your method allows for larger receptive field sizes, but because of the increased weight sharing further from the center point, this comes at the trade-off of lower resolution information at higher distances correct? Could you touch upon this?
4. In this same line, I would like to see the authors more explicitly expand on what they think ultimately results in the performance increase of their method over conventional CNNs? Do you think it is a result of the increased receptive field, the way in which you treat local and nonlocal information at different resolutions, both?
5. Also, can you touch upon some use-cases in which this assumption of locality necessarily does not hold, but for which conventional CNNs are still used in practice? For example, how would your method perform on speech speech signals, which are generally sampled at very high rates and may exhibit very non-local patterns?

**Post-rebuttal response**
I would like to thank the authors for their extensive responses to my questions, and the questions of other reviewers. I appreciate the fact that you included a discussion for multiple of the concerns and questions raised by me and other reviewers in your manuscript, I feel this improves the quality of the work.

I noticed that my concern regarding the motivation for your specific approach to weight sharing in convolution kernels was shared by other reviewers as well. I think you greatly improved your motivation by including an analysis on pixel statistics.

The additional experiments and comparisons made by the authors in their revision strengthen this submission further, and show the relevance of this approach by showing it performs even when compared to more recent approaches.

The authors additionally addressed a number of limitations of their work in their revision, which make the work more transparent.

In light of these improvements, I am slightly raising my recommendation. I thank the authors for an interesting submission!


**Limitations:**

Authors briefly discuss the limitations of their work; it adds hyperparameters and isn’t very memory efficient. What I would like the authors to touch upon are limitations which may be present in their method inherently, as a result of using the log-polar space to define convolution kernels (see also my questions above).


**Strengths And Weaknesses:**

The authors give a very clear overview of the relevance and motivation behind their method, by first highlighting limitations of current approaches to larger ERFs and from there introducing their approach to address these issues. As far as I know, their work is highly original.

Section 3.1 contains a comprehensive overview of the method, although I get the sense that an illustration visually explaining your method (i.e. show how R relates to the kernel size, how $L_{\theta}, L_r$ change the kernel), as you did for initial angle and eccentricity in fig 2, could be very helpful in getting the specifics across. In general, the paper is written in a clear manner, but the number of inline equations and parameters make it somewhat hard to read at points.

The choice of experiments is motivated well; the goal of the authors is to make a direct comparison between conventional convolutional layers and the introduced log-polar space convolutions. The experiments themselves seem to be chosen fairly for this goal, and decidedly show the improvement of LPSCs over conventional convolutions.

I appreciate that the authors chose to investigate how their framework could be implemented through highly optimised conventional convolutions. This allows for a much more direct comparison with conventional convolutions.

Misc:  Under fig 2a and in line 171, 187 “is slide” -> “is slid”.

---

> ### Author Response · Authors · 2022-08-02
> **Added discussions on use-cases of ellipsoid LPSCs, the conditions, assumptions, and limitations of using LPSCs, the use-cases where the assumption does not hold, and added ablation studies on the improvement (1/3)**
>
> We thank the reviewer for the valuable comments.
>
> **Q1.** Do you have an intuition for when you would want to use ellipsoid LPSCs? Are there any specific use-cases that come to mind?
>
> **A1.** Thank you for this enlightening comment. In natural images, different objects or local parts have different shapes and orientations, so it may be suitable to use ellipsoid LPSC kernels to fit objects or parts with different orientations and sizes. For instance, in image understanding, we can fuse multiple ellipsoid LPSC kernels with different sizes (local receptive fields) and orientations in a multi-channel parallel manner. Moreover, a given ellipsoid LPSC kernel can be scaled without changing the parameters. If the orientations and size scaling of LPSC kernels can be learned while learning the convolution parameters, rotation and scale invariant features can be extracted from objects or local parts, so that the learned representations are more robust. We intend to explore these directions in our future work. We have added this discussion in Appendix A.5, Page 21.
>
>
> **Q2.** Your method actually gives a definition for kernel values over the entire continuous input space. Does this mean your method could be readily applied to irregularly sampled data? Are there any specifics of your method that would inhibit you from doing this? Have you attempted this?
>
> **A2.** Thank you for this inspiring comment. As long as the data distribution conforms to our assumption, i.e., in local areas, the data sample closer to the center point is more correlated with it, and the correlation decays rapidly with the increase of their distance, our LPSC can also be applied to irregularly sampled data, provided that the relative distances and angles between sampled data points are defined. However, if the mask matrix to indicate the region indexes of positions cannot be pre-computed as a lookup for irregularly sampled data, the speed of LPSC will be very slow, because the region that each sampled data falls in should be calculated on-the-fly. We have not yet attempted to apply LPSC to irregularly sampled data. In the future, we intend to apply LPSC to graph data and 3D point cloud data. We have added part of this discussion in limitations of our method in Section 3.4, Page 6.

---

> > ### Author Response · Authors · 2022-08-02
> > **Added discussions on use-cases of ellipsoid LPSCs, the conditions, assumptions, and limitations of using LPSCs, the use-cases where the assumption does not hold, and added ablation studies on the improvement (2/3)**
> >
> > **Q3.** The authors motivate their log-polar space kernel definition by saying that in natural images often correlation between local pixels is higher than correlation between more distant pixels and therefore we can assign the same parameters to larger patches of pixels further away. I don’t think this argument necessarily makes sense; indeed natural images are locally correlated, but I believe this only says that local pixels should be assigned larger weights values, not that larger patches of pixels further away should be assigned identical weight values. Indeed, your method allows for larger receptive field sizes, but because of the increased weight sharing further from the center point, this comes at the trade-off of lower resolution information at higher distances correct? Could you touch upon this?
> >
> > **A3.** Thank you for pointing out this. We have calculated the correlations between image pixels in different log-polar regions and the center pixels on the training set of CIFAR-100. Specifically, for each pixel in each image, we divide its $11 \\times 11$ neighboring area into different regions by LPSC with 3 distance levels, 8 direction levels, and a growth rate of 2. The center pixels of all areas form the center set. The pixels at the same position of all areas also form a pixel set. For each position, we calculate the correlation score between the corresponding pixel set and the center set. The correlation scores of positions in the same region of all training images are averaged to obtain the correlation score between the region and the center pixel. In this way, we obtain the correlation scores from all $3 \\times 8$ regions to the center pixel, as shown in the table below. In further regions, the average correlation of pixels is lower, and the average correlations within different regions of the same distance level are comparable. The farther the region is from the center point, the lower the correlation, and the decay is rapid (approximately exponentially decreasing). We have added this motivation in Section 1 and the statistics in Appendix A.1.
> >
> > Table 1. Correlation scores between different regions and the center pixel.
> > | Direction level | 4 | 3 | 2 | 1 | 5 | 6 | 7 | 8 |
> > | :------| :----| :----| :----| :----| :----| :----| :----| :----|
> > | Distance level 1 | 0.6795 | 0.8268 | 0.6794 | 0.9162 | 0.8323 | 0.6799 | 0.8277 | 0.6800 |
> > | Distance level 2 | 0.4663 | 0.5844 | 0.4620 | 0.5967 | 0.5967 | 0.4620 | 0.5852 | 0.4665 |
> > | Distance level 3 | 0.2113 | 0.2779 | 0.1992 | 0.2983 | 0.2976 | 0.1968 | 0.2764 | 0.2097 |
> >
> > Indeed, these results can only explicitly support that local pixels should be assigned larger weight values, not that larger patches of pixels further away should be assigned identical weight values. To reduce the number of parameters while utilizing all the pixels in the receptive field, we need to perform parameter sharing among pixels. Neighboring pixels are more important so fewer pixels share the same weight in LPSC. On the one hand, pixels in faraway regions are less correlated to the center point. On the other hand, considering that those faraway points in the same region are also locally adjacent, the correlations among them are usually strong. Therefore, using more pixels in larger faraway regions for parameter-sharing is more reasonable and can better alleviate the loss of lower resolution information. A faraway region as a whole interacts with the center point. Nevertheless, parameter sharing in LPSC aims to expand the local receptive field without increasing the number of parameters, but the cost is the loss of some fine-grained information at higher distances since the resolution is blurred, as pointed out by the reviewer. Our LPSC achieves a balance and may be better suitable when no sufficient training data is available where fewer parameters are preferred and when the model is shallow where larger LRFs in lower layers are more crucial. This is also consistent with our experimental results. We have added part of this discussion in limitations of our method in Section 3.4, Page 6.

---

> > > ### Author Response · Authors · 2022-08-02
> > > **Added discussions on use-cases of ellipsoid LPSCs, the conditions, assumptions, and limitations of using LPSCs, the use-cases where the assumption does not hold, and added ablation studies on the improvement (3/3)**
> > >
> > > **Q4.** In this same line, I would like to see the authors more explicitly expand on what they think ultimately results in the performance increase of their method over conventional CNNs? Do you think it is a result of the increased receptive field, the way in which you treat local and nonlocal information at different resolutions, both?
> > >
> > > **A4.** Thank you for this suggestion. As discussed in A3, our method balances the number of parameters for increased receptive fields by using the special configuration and pooling in the log-polar space to compensate for resolution reduction. As shown in Tab .1, Tab.9-Tab.10, Appendix A.3, compared with conventional convolution, circle convolution, and dilated convolution with comparable LRFs and number of parameters, our LPSC achieves better results. Dilated convolution also outperforms conventional convolution in segmentation. These results show the effectiveness of increasing receptive fields in a suitable way. To further verity whether the improvement is fully due to the increased receptive fields without increasing the number of parameters (i.e., the effectiveness of our way to treat local and nonlocal information), we compare with the square-space-pooling-based convolution. Specifically, we averagely divide the kernel into different square regions, e.g., a $9 \\times 9$ kernel can be divided into $3 \\times 3$ regions with a size of $3 \\times 3$. All positions in the same square region share the same parameter. We denote this alternative convolution method by square convolution, which also increases LRF with fewer parameters. Comparisons with square convolution are shown as follows, where we perform square convolution once since its performance is lower than other methods by a margin. Our LPSC also outperforms the average square convolution with different hyper-parameters, as presented in Appendix A.3. This indicates that the spatial structure designed in the log-polar space can better capture contextual information. These results and discussions are included in Tab. 9 and Tab. 10 in Appendix A.3. In summary, we believe that the improvement is due to the increased LRF by our way to treat local and non-local information in the log-polar space. LPSC increases LRFs by regularizing the low-resolution of faraway continuous regions without increasing the number of parameters.
> > >
> > > Table 6. Comparison of different convolution methods with AlexNet.
> > > | Dataset | Square | LPSC |
> > > | :------| :----| :----|
> > > | CIFAR-10 | 76.10 | **78.44** (0.12) |
> > > | CIFAR-100 | 44.50 | **47.43** (0.20) |
> > >
> > > Table 7. Comparison of different convolution methods with VGG-19.
> > > | Dataset | Square | LPSC |
> > > | :------| :----| :----|
> > > | CIFAR-10 | 90.01 | **93.92** (0.06) |
> > > | CIFAR-100 | 66.64 | **73.13** (0.12) |
> > >
> > > Table 8. Comparison of different convolution methods with ResNet-20.
> > > | Dataset | Square | LPSC |
> > > | :------| :----| :----|
> > > | CIFAR-10 | 89.84 | **91.81** (0.21) |
> > > | CIFAR-100 | 63.54 | **67.63** (0.27) |
> > >
> > >
> > >
> > > **Q5.** Also, can you touch upon some use-cases in which this assumption of locality necessarily does not hold, but for which conventional CNNs are still used in practice? For example, how would your method perform on speech speech signals, which are generally sampled at very high rates and may exhibit very non-local patterns?
> > >
> > > **A5.** Thank you for this insightful comment. Conventional convolution can also be applied to semantically dense data such as speech signals, text sequences, and amino acid sequences. Each unit or sampling point is relatively independent and has its own unique semantics, which is different from the semantics of its adjacent sampling points. For these data, even if points in regions far from the center point are also not suitable for sharing parameters, so if LPSC is applied to such data, fine-grained information will be blurred and lost. The proposed LPSC is more suitable for semantically sparse visual data that contains a large amount of redundant information. The region-division configuration in the log-polar space is based on the locality assumption to assign larger weights for neighboring regions, and the parameter-sharing within regions can efficiently model redundant data. We have added part of this discussion Section 3.4, Page 6.
> > >
> > > **Q6.** Misc:  Under fig 2a and in line 171, 187 “is slide” -> “is slid”.
> > >
> > > **A6.** Thank you for pointing them out. We have revised the typos.

---

> > > > ### Author Response · Authors · 2022-08-08
> > > > **Follow-up on the responses and revised manuscript**
> > > >
> > > > Dear Reviewer AZ8Z,
> > > >
> > > > Thank you again for your valuable comments. Since the discussion stage is closing soon, we would be grateful if you could let us know whether our responses and revised manuscript have addressed your concerns and whether there are further comments.
> > > >
> > > > Sincerely,
> > > > Authors

---

> > > > > ### Comment · Reviewer_AZ8Z · 2022-08-08
> > > > > **Post-rebuttal response**
> > > > >
> > > > > I've updated my review. Thank you for your thorough rebuttal!

---

> > > > > > ### Author Response · Authors · 2022-08-08
> > > > > > **Thanks again**
> > > > > >
> > > > > > Thank you very much again for your valuable and constructive comments, which we believe have improved our manuscript significantly.

---

### Official Review · Reviewer_4RP8 · 2022-07-06

**Rating:** 5
**Confidence:** 4
**Soundness:** 2 fair
**Presentation:** 2 fair
**Contribution:** 3 good

**Summary:**

# Summary
In this paper, an elliptic convolution kernel is proposed, and its local receptive field is adaptively divided into different regions according to the relative direction and logarithmic distance


**Questions:**

1.In Table 1, the accuracy of using alexnet in cifar10 and cifar100 is only about 77 and 45 respectively. Although it is improved compared with other convolution cores, it is also difficult to be convincing. At the same time, the improvement on vgg19 and resnet20 is too subtle to verify the effectiveness of the convolution kernel.More experiments are needed to verify the validity of the statements in the paper.

2.Why choose resnet18 instead of resnet50 for experiments on Imagenet.

3.In Table 2 and table 3,Why not use dilated convolution with your method.

4.As mentioned in Section 4.1 of the paper, the advantages of LPSC are weakened for deeper models. Please explain why.

5.More experiments are needed to prove that LPSC can obtain more effect of improving receptive field than dilated convolution. At the same time, improving receptive field is mainly used in semantic segmentation, so more experiments on semantic segmentation are needed.

**Ethics Review Area:**

["I don’t know"]

**Limitations:**

The idea is very good, but I need better experimental results to support it.

**Strengths And Weaknesses:**

# Strengths
1.The proposed method is very novel
# Weaknesses
1.The comparison objects selected in the comparison experiment are very old, and the performance has not been greatly improved.

2.Poor effect in deep network model.

3.Insufficient experiments in image segmentation.

---

> ### Author Response · Authors · 2022-08-02
> **Added comparisons with more recent methods, explanation on the effect in deep models, and added comparisons with dilated convolution and experiments in segmentation (1/3)**
>
> We thank the reviewer for the valuable comments.
>
> **Q1.** The comparison objects selected in the comparison experiment are very old, and the performance has not been greatly improved.
>
> **A1.** Thank you for pointing out this. We aim to design a new convolutional layer for convolutional neural networks that can enlarge the local receptive field (LRF) without increasing the amount of computation and parameters, so we compare with convolution kernels with the same target. We have also added experimental comparisons with the circle convolution ([40] in the revised manuscript) and the deformable convolution [6] (we use the Pytorch implementation in https://github.com/oeway/pytorch-deform-conv and perform once since the accuracy is lower than other methods by a margin) in Appendix A.3, Tab. 9-Tab. 10. The results are also shown below. For circle convolution with AlexNet, we have tried several different kinds of initializations but all the trained networks collapsed into a trivial solution. Our LPSC outperforms [A] and the deformable convolution with fewer or comparable parameters. We have added these experiments in Appendix A.3 of the revised manuscript.
>
> Table 1. Comparison of different convolution methods with AlexNet.
> | Dataset | Ori | Dilation | Deformable | Circle | LPSC |
> | :------| :----| :----| :----| :----| :----|
> | # Params (M) | 2.47 | 2.34 | 2.47 | 2.46 | 2.31 |
> | CIFAR-10 | 77.43 (0.25) | 75.42 (0.06) | 75.98 | 10 | **78.44** (0.12) |
> | CIFAR-100 | 43.98 (0.43) | 44.43 (0.10) | 41.96 | 1 | **47.43** (0.20) |
>
> Table 2. Comparison of different convolution methods with VGG-19.
> | Dataset | Ori | Dilation | Deformable | Circle | LPSC |
> | :------| :----| :----| :----| :----| :----|
> | # Params (M) | 20.04 | 20.08 | 20.04 | 20.08 | 20.08 |
> | CIFAR-10 | 93.54 (0.06) | 93.46 (0.14) | 92.53 | 93.85 (0.08) | **93.92** (0.06) |
> | CIFAR-100 | 72.41 (0.17) | 73.03 (0.34) | 69.32 | 72.76 (0.28) | **73.13** (0.12) |
>
> Table 3. Comparison of different convolution methods with ResNet-20.
> | Dataset | Ori | Dilation | Deformable | Circle | LPSC | LPSC-CC |
> | :------| :----| :----| :----| :----| :----| :----|
> | \# Params (M) | 0.27 | 0.27 | 0.27 | 0.27 | 0.27 | 0.27 |
> | CIFAR-10 | 91.66 (0.13) | 91.44 (0.10) | 90.27 | 91.76 (0.14) | **91.81** (0.21) | **92.01** (0.08) |
> | CIFAR-100 | 67.56 (0.27) | 66.90 (0.25) | 65.44 | 67.58 (0.28) | **67.63** (0.27) | **68.09** (0.27) |
>
> In summary, our LPSC improves different convolutional networks on different datasets in different tasks, ranging from AlexNet to DeepLabV3+, from classification to segmentation. The improvements are obtained without changing any training hyperparameters (learning rate, etc) based on the average results of multiple runs. LPSC even achieves significant improvements in some cases. Moreover, compared with conventional convolution, LPSC has a larger LRF with comparable parameters, or has much fewer parameters with comparable LRF. We believe that these results already demonstrate that LPSC is an effective and versatile alternative or supplement to conventional convolution for increasing the LRF.
>
> [A] Kun He, Chao Li, Yixiao Yang, Gao Huang, and John E. Hopcroft. Integrating large circular kernels into cnns through neural architecture search. arXiv preprint arXiv:2107.02451, 2021.
>
>
> **Q2.** Poor effect in deep network model.
>
> **A2.** When the model goes deeper, the advantage of using LPSC in the lower layers is weakened. The reason may be that stacking deeper layers (with residual connections) can achieve large LRF using multi-layer small regular kernels, which eases the need for large LRF in lower layers. Moreover, the non-uniform distribution of parameters in LPSC may cause information dispersion and over-smoothing, therefore it is more difficult to model residuals. Our proposed cross-convolution strategy (presented in Section 3.3) can alleviate this problem by replacing part of conventional convolutions with LPSC in lower layers. As shown in Tab. 2, with this strategy, LPSC-CC consistently achieves improvement with the deep ResNet-110 backbone. We have included the explanation on the limited improvement in deeper networks in Lines 698-701, Appendix A.3.

---

> > ### Author Response · Authors · 2022-08-02
> > **Added comparisons with more recent methods, explanation on the effect in deep models, and added comparisons with dilated convolution and experiments in segmentation (2/3)**
> >
> > **Q3.** Insufficient experiments in image segmentation.
> >
> > **A3.** All experiments in this paper are conducted using a single GPU, but performing image segmentation on larger datasets such as MS COCO, Citypscapes, and ADE20K often requires more than 4 GPUs, which is beyond the capacity of our computing resources. Our experiments on VOC 2012 and DRIVE at least show the effectiveness of our LPSC with relatively-small backbones and light-weight segmentation models for both natural images and medical images. We have added experiments on the VOC 2012 dataset with the cross-convolution strategy, where 1/4 of the dilated convolution kernels with the largest rate in ASPP of DeepLabv3+ are replaced by LPSC kernels with the largest preset kernel size ($11\\times 11, (L_r, L_\\theta, g) =(3,8,2) $). Results are shown as follows. A small part of LPSC kernels in the cross-convolution strategy can also improve the performance.
> >
> > Table 4. Results on segmentation.
> > | Method | oAcc | mAcc | fAcc | mIoU |
> > | :------| :----| :----| :----| :----|
> > | DeepLabv3+$^{*}$ | 0.9230 | 0.8332 | 0.8652 | 0.7144 |
> > | DeepLabv3+LPSC-CC | 0.9252 | 0.8313 | 0.8678 | 0.7191 |
> > | DeepLabv3+LPSC | 0.9273 | 0.8388 | 0.8714 | 0.7260 |
> >
> >
> > **Q4.** In Table 1, the accuracy of using alexnet in cifar10 and cifar100 is only about 77 and 45 respectively. Although it is improved compared with other convolution cores, it is also difficult to be convincing. At the same time, the improvement on vgg19 and resnet20 is too subtle to verify the effectiveness of the convolution kernel. More experiments are needed to verify the validity of the statements in the paper.
> >
> > **A4.** For our experiments on image classification, we use the publicly available implementations at https://github.com/bearpaw/pytorch-classification as our baselines. The results reported there do not include standard deviations, so we re-run different models multiple times to calculate the means and standard deviations. Our reproduced results are consistent with the reported results, including the results of AlexNet, VGG, and ResNet. The improvements of our LPSC are obtained without changing any training hyperparameters (learning rate, etc) based on the average results of multiple runs. Although the improvement in some cases does not seem large, our LPSC achieves general improvement across different network architectures, different datasets, and different tasks. Our cross-convolution strategy achieves further improvement for deeper models such as ResNet-110, as shown in Tab. 2.
> >
> >
> > **Q5.** Why choose resnet18 instead of resnet50 for experiments on Imagenet.
> >
> > **A5.** In the ReadMe of the baseline code (https://github.com/bearpaw/pytorch-classification), only results with ResNet18 and ResNeXt-50 (32x4d) are reported. Following the example given in the code, we choose ResNet18 so that we can directly compare with the reported experimental results. With a single GPU, it takes over a week to perform ResNet-18 on ImageNet and it will be slower with a larger network like ResNet50. Instead, we have performed experiments with ResNet-110 on the CIFAR-100 dataset to demonstrate the effectiveness of LPSC-CC on deeper networks under limited computing resources, as shown in Tab. 2 of the revised manuscript.

---

> > > ### Author Response · Authors · 2022-08-02
> > > **Added comparisons with more recent methods, explanation on the effect in deep models, and added comparisons with dilated convolution and experiments in segmentation (3/3)**
> > >
> > > **Q6.** In Table 2 and table 3, Why not use dilated convolution with your method.
> > >
> > > **A6.** Thank you for this suggestion. We have added experimental comparisons with dilated convolution in ResNet-110 in Table 2. Specifically, we replace the first convolution layer with dilated convolution with a dilation rate of 2. As shown below, LPSC outperforms dilated convolution with the same LRF. LPSC-CC further improves the performance. In Table 3, performing dilated convolution on ImageNet is very slow within one GPU and we will supplement the experimental results after the experiment has been done. Moreover, we have included results of dilated convolution with different hyper-parameters in Tab. 9 in Appendix A.3. Moreover, in segmentation experiments, we replace the dilated convolution in some layers with our LPSC to better employ larger LRF. We believe that these results prove that LPSC can obtain more effect on improving receptive field than dilated convolution.
> > >
> > > Table 5. Results with ResNet-110 on CIFAR-100.
> > > | Model | Time | Acc |
> > > | :------| :----| :----|
> > > | Conv | 0.025 (0.0053) | 73.50 (0.24) |
> > > | Dilation | 0.025 (0.0041) | 73.75 (0.48) |
> > > | LPSC | 0.036 (0.0047) | 73.54 (0.38) |
> > > | LPSC-CC | 0.135 (0.0035) | **74.13** (0.14) |
> > >
> > >
> > > **Q7.** As mentioned in Section 4.1 of the paper, the advantages of LPSC are weakened for deeper models. Please explain why.
> > >
> > > **A7.** Please refer to A2.
> > >
> > >
> > > **Q8.** More experiments are needed to prove that LPSC can obtain more effect of improving receptive field than dilated convolution. At the same time, improving receptive field is mainly used in semantic segmentation, so more experiments on semantic segmentation are needed.
> > >
> > > **A8.** Thank you for the suggestions. Please refer to A6 and A4 for additional experiments for dilated convolution and semantic segmentation. Please refer to Appendix A.3 for a more detailed comparison with other convolution methods to demonstrate the effectiveness of our LPSC.

---

> > > > ### Author Response · Authors · 2022-08-08
> > > > **Follow-up on the responses and revised manuscript**
> > > >
> > > > Dear Reviewer 4RP8,
> > > >
> > > > Thank you again for your valuable comments. Since the discussion stage is closing soon, we would be grateful if you could let us know whether our responses and revised manuscript have addressed your concerns and whether there are further comments.
> > > >
> > > > Sincerely,
> > > > Authors

---

> > > > > ### Comment · Reviewer_4RP8 · 2022-08-10
> > > > > **Post-rebuttal response**
> > > > >
> > > > > I have updated my rating, thanks for your rebuttal

---

> > > > > > ### Author Response · Authors · 2022-08-10
> > > > > > **Thanks again**
> > > > > >
> > > > > > Thank you very much again for your valuable and constructive comments, which we believe have improved our manuscript significantly.

---

### Official Review · Reviewer_aBNv · 2022-07-08

**Rating:** 5
**Confidence:** 4
**Soundness:** 3 good
**Presentation:** 3 good
**Contribution:** 2 fair

**Summary:**

This paper introduces a new convolution operator. Instead of using a rectangle convolution kernel following the grid structure of the input data, the authors propose to define convolution kernels in a log-polar space. Specifically, the kernel weight depends on the distance and direction wrt to the center, and the density of weights is inversely proportional to the distance to the center. Because the weights are not defined on the same regular grid as the input data, a pooling operation is performed before applying the convolution kernel. The main benefit of the proposed approach is that less parameters are needed when the size of the receptive field increases, based on the insight that pixels adjacent to the center should have more contributions to the output. Empirical results on multiple tasks and models show that the proposed method outperforms models using only standard convolution kernels.

**Questions:**

1. Given that the proposed method introduces additional computation and parameters, the overhead should be considered in the evaluation, e.g. compare overhead vs accuracy instead of single point accuracy
2. Validation should be use to determine the meta-parameters for a fair comparison


**Limitations:**

1. There doesn't seem to be any obvious negative social impact for this work.
2. The authors describe some of the limitations of the proposed method, although some additional information may help understanding the limitations, e.g. the exact memory overhead.

**Strengths And Weaknesses:**

* Strengths
  * The key idea is intuitive and well motivated
  * The proposed approach is generic and is applicable to most existing CNNs, and it is easy to implement
  * The results suggest that the proposed convolution operator consistently perform better than standard convolution
* Weakness
  * The method requires additional computation and parameter during inference time, and the experiment does not really perform apple to apple comparisons
  * The accuracy improvement is not significant consider that it introduces additional computation and memory overhead
  * Using the same setup does not imply fair comparison. One should optimize each model independently for a fair comparison.
  * The proposed method introduces additional meta-parameters, which are determined by the accuracy on the test set according to L281 and leads to unfair comparison

* After rebuttal
  * The concerns regarding the accuracy are properly addressed, i.e. training setup and meta-parameters
  * The claim for lower computational cost is still not clearly explained. L222 assumes standard convolution and LPSC has comparable M and N. But in practice, most standard convolution has M=N=1, while LPSC has M and N > 1. Therefore, it's not clear why the flops is lower when 3x3 kernels are used.

---

> ### Author Response · Authors · 2022-08-02
> **Clarification on the computation complexity, parameters, hyper-parameter selection, and fair comparison, and evaluations on time and FLOPs (1/2)**
>
> We thank the reviewer for the valuable comments.
>
> **Q1.** The method requires additional computation and parameter during inference time, and the experiment does not really perform apple to apple comparisons
>
> **A1.** We respectfully remind the reviewer that our method does not require additional parameters or computation than conventional convolution. In fact, given the same LRF (kernel size), our LPSC kernel has much fewer parameters compared with the conventional convolution kernel, due to the parameter sharing within regions, as clearly presented in line 161-169, Section 3.1, Page 4. For computations, our LPSC requires the same addition operations and much fewer multiplications, as clearly analyzed in “complexity” in Section 3.4, Page 5. This is also verified by the computational comparison of FLOPs in the experiments in Section 4.1, Page 7, where our method has much lower FLOPs than other convolution methods. For apple-to-apple comparisons, we simply follow the experimental setups turned for conventional convolutions. To perform our LPSC. We directly replace conventional convolutions in the lower layers. We compare with dilation convolutions using the most fitted hyper-parameters, where the results of dilation convolution with different hyper-parameters are presented in Tab. 9 in Appendix A.2.
>
>
> **Q2.** The accuracy improvement is not significant consider that it introduces additional computation and memory overhead
>
> **A2.** Our method does not introduce additional computation, as discussed in A.1. We respectively remind the reviewer that our LPSC improves different convolutional networks on different datasets in different tasks, ranging from AlexNet to DeepLabV3+, from classification to segmentation. The improvements are obtained without changing any training hyperparameters (learning rate, etc) based on the average results of multiple runs. LPSC even achieves significant improvements in some cases. Moreover, compared with conventional convolution, LPSC has a larger LRF with comparable parameters, or has much fewer parameters with comparable LRF. We believe that these results already demonstrate that LPSC is an effective and versatile alternative or supplement to conventional convolution for increasing the LRF.
>
>
> **Q3.** Using the same setup does not imply fair comparison. One should optimize each model independently for a fair comparison.
>
> **A3.** Please note that on all experiments, we directly used different provided codes available at Github for training and evaluation and did not tune or change any provided hyper-parameters (unless on ImageNet, we reduce the batch size and learning rate by 4 times due to the memory constraint). These hyper-parameters are tuned for the original networks. Tuning them for our LPSC will increase the advantage of our method. In fact, the influences of these hyper-parameters are well known and the settings used in these implementations are quite common in the literature. Therefore, fixing them to these common values is widely adopted for comparison in many previous works.
>
> **Q4.** The proposed method introduces additional meta-parameters, which are determined by the accuracy on the test set according to L281 and leads to unfair comparison
>
> **A4.** In L281, based on the ablation study on hyper-parameters of LPSC in A.2, for $5 \\times 5$ LPSC kernels, we fix $(L_r, L_\\theta, g)$ to (2, 6, 2) on all experiments (in this case (2, 6, 2) is equivalent to (2, 6, 3) as explained in footnote 7, Page 14). On CIFAR10/100 (with VGGNet and ResNet) and ImageNet, only $5 \\times 5$ LPSC kernels are used. For AlexNet, the first convolution layer uses $11 \\times 11$ kernels, and we set $(L_r, L_\\theta, g)$ to (3, 8, 2), where large $L_r$ and $L_\\theta$ are used to reduce the gap to the number of parameters of original conventional convolution kernels, but $L_r$ and $L_\\theta$ cannot be larger otherwise less than one pixel will fall in some innermost regions. On the two segmentation datasets, when the kernel size is larger than $5 \\times 5$, the hyper-parameters of LPSCs are set so that LPSCs have comparable local receptive fields and number of parameters with the original dilated convolutions in the corresponding architectures. E.g., on the DRIVE dataset, in addition to the common setting of (2, 6, 2), we also try larger kernels with a $(L_r, L_\\theta, g)$ of (3, 8, 1.5) to increase the number of parameters, which results in comparable results with (2, 6, 2). Therefore, after ablation study, the hyper-parameters of LPSC are set heuristically on all other datasets rather than determined by the accuracy on the test set. For $5 \\times 5$ kernels, we fix $(L_r, L_\\theta, g)$ to (2, 6, 2). For larger kernels but less than $11$, we increase $ L_\\theta $ to 8. For $11 \\times 11$ kernels, we further increase $L_r$ to 3. Further tuning these hyper-parameters based on a validation set may further improve the performance of our LPSC.

---

> > ### Author Response · Authors · 2022-08-02
> > **Clarification on the computation complexity, parameters, hyper-parameter selection, and fair comparison, and evaluations on time and FLOPs (2/2)**
> >
> > **Q5.** Given that the proposed method introduces additional computation and parameters, the overhead should be considered in the evaluation, e.g. compare overhead vs accuracy instead of single point accuracy
> >
> > **A5.** Thanks for this suggestion. On the CIFAR10 dataset, the training time for one epoch and the testing time for AlexNet are 3.4808 and 0.9083, respectively; after replacing conventional convolutions with LPSCs in the first two layers, the training time for one epoch and the testing time are 23.3981 and 3.7476, respectively. The inference times per epoch of using different convolutions for ResNet-110 are compared in Tab. In our implementation, LPSC runs much slower than conventional convolution and requires much memory overhead (the complexity analysis on memory is presented in Section 3.4), but this is because we use of-the-shell conventional convolution modules to implement LPSC. To this end, we must first apply log-polar space pooling with the fold and unfold operations in Pytorch, which consume much time and space complexity. However, since the mask indicating the division of regions is pre-calculated once in the constructor when instantiating the network class, it involves $(2M+1) \\times (2N+1)$ lookups but requires ignorable additional computations and spaces. From Eq. (1) and Eq.(2), for a $(2M+1) \\times (2N+1) \\times C$ kernel, conventional convolution involves $(2M+1) \\times (2N+1) \\times C$ multiplications and $(2M+1) \\times (2N+1) \\times C$ additions, while LPSC with $L_r$ distance levels and $L_\\theta$ direction levels only involves $2*L_r \\times L_\\theta \\times C$ multiplications and $(2M+1) \\times (2N+1) \\times C$ additions. Typically, if $L_r$ is set to 2, $L_\\theta$ is set to 6, only $24C$ multiplications are executed, however, even for a small $(2M+1) \\times (2N+1) = 5 \\times 5$ kernel, conventional convolution executes 25C multiplications. For a $7 \\times 7$ kernel, multiplications increase to $49C$. Since LPSC kernels have much fewer parameters than conventional convolution kernels, fewer gradient calculations are required in back-propagation. Therefore, theoretically, LPSC has smaller forward computational and training complexities. This is the reason why LPSC has fewer FLOPs. On the CIFAR dataset with AlexNet, the FLOPs (recorded by the fvcore toolbox) of conventional convolution, dilated convolution, deformable convolution, squared convolution, and LPSC (w.r.t. the best performances in Tab. 3(a)) are 14.95M, 24.71M, 15.12M, 13.77M, and 11.42M, respectively. LPSC can be greatly accelerated if it is directly implemented with CUDA or by directly adapting the underlying code of convolutions in the integrated framework. The number of parameters has been compared in Tab.1, Tab. 3, Tab. 7, Tab. 9, and Tab. 10. We have included the comparisons on the training/inference time and FLOPs in Section 4 (e.g., Tab. 2, the end of Page 7) and Appendix A.2 (the end of Page 16) of the revised manuscript.
> >
> >
> > **Q6.** Validation should be use to determine the meta-parameters for a fair comparison
> >
> > **A6.** Please refer to Q4.

---

> > > ### Author Response · Authors · 2022-08-08
> > > **Follow-up on the responses and revised manuscript**
> > >
> > > Dear Reviewer aBNv,
> > >
> > > Thank you again for your valuable comments. Since the discussion stage is closing soon, we would be grateful if you could let us know whether our responses and revised manuscript have addressed your concerns and whether there are further comments.
> > >
> > > Sincerely,
> > > Authors

---

### Official Review · Reviewer_aXhX · 2022-07-09

**Rating:** 5
**Confidence:** 4
**Soundness:** 2 fair
**Presentation:** 2 fair
**Contribution:** 2 fair

**Summary:**

This paper presents a novel variant of convolutional layer (log-polar space convolution LPSC) in neural networks, wherein the convolution kernel is not rectangular, but rather elliptical with different sized regions, which are pooled. It also describes a log-polar space pooling for convenient implementation. Finally, it applies the proposed method to various neural network architectures and tasks thereof.

**Questions:**

Question 1: Although only a bit earlier work, the authors should discuss the relation to this paper : He, Kun, et al. "Integrating Large Circular Kernels into CNNs through Neural Architecture Search." arXiv preprint arXiv:2107.02451 (2021).
That work also cites other papers that should be looked at (some of these are already cited by the current paper): “... the deformable convolution (Dai et al., 2017; Zhu et al., 2019) … Similarly, the deformable kernel (Gao et al., 2020) …including quasi-hexagonal convolution (Sun et al., 2016), blind-spot convolution (Krull et al., 2019), asymmetric convolution (Ding et al., 2019), etc.”

Question 2: Have the authors studied how much performance comes from the polar coordinates and how much from the growing size of areas that are pooled (further areas are pooled more). Could the pooling be done in the normal rectangular grid and how much would be gained by that.


**Limitations:**

The authors properly note the drawbacks of additional hyperparameters and memory overhead. For other limitations, see above for weaknesses and questions in this review.

**Strengths And Weaknesses:**

Strengths:
--------------

Strength 1: The idea of log-polar convolution makes intuitively sense since the rectangular pixel quantization is more an artifact of the capture and storage systems rather than a feature of nature.

Strength 2: A practical pooling method for easily incorporating the method into existing SW, HW and networks.

Weaknesses:
------------------

Weakness 1: There are other non-typical kernel shapes that have been proposed in the literature (some are cited in the paper). The authors should compare their method to the best of the previously published methods.

Weakness 2: The method does not seem to bring a huge gain, but result into slower training and inference and additional hyperparameters. While this should still be ok for data-limited setup because of the potential regularization effect of the proposed system, it is not clear whether it is better to use the proposed method or just a somewhat larger CNN to achieve similar gains. Please study the regularization / generalization effect of the proposed method further.

Weakness 3: The motivation of the log-polar space convolution would be better if natural image statistics were used to justify it. Now it stands out as rather ad-hoc method.

After the rebuttal
---------------------
I have reviewed the author feedback and the they have done a good job in clarifying their work. I have reflected this in the score.

---

> ### Author Response · Authors · 2022-08-02
> **Added comparisons with more recent methods, explanation on the regularization effect, added image statistics for justification, and added ablation studies (1/3)**
>
> We thank the reviewer for the valuable comments.
>
> **Q1**: Weakness 1: There are other non-typical kernel shapes that have been proposed in the literature (some are cited in the paper). The authors should compare their method to the best of the previously published methods.
>
> **A1:** Thank you for pointing out them. We have added discussions with circle convolution [A] ([40] in the revised manuscript) and some references there in Section 2 in the revised version, page 3. We have also added experimental comparisons with [A] and the deformable convolution (we use the Pytorch implementation in https://github.com/oeway/pytorch-deform-conv and perform once since the accuracy is lower than other methods by a margin) in Appendix A.3, Tab. 9-Tab. 10. The results are also shown below. For circle convolution [A] with AlexNet, we have tried several different kinds of initializations but all the trained networks collapsed into a trivial solution. Our LPSC outperforms [A] and the deformable convolution with fewer or comparable parameters. We have added these experiments in Appendix A.3 of the revised manuscript.
>
> Table 1. Comparison of different convolution methods with AlexNet.
> | Dataset | Ori | Dilation | Deformable | Circle | LPSC |
> | :------| :----| :----| :----| :----| :----|
> | # Params (M) | 2.47 | 2.34 | 2.47 | 2.46 | 2.31 |
> | CIFAR-10 | 77.43 (0.25) | 75.42 (0.06) | 75.98 | 10 | **78.44** (0.12) |
> | CIFAR-100 | 43.98 (0.43) | 44.43 (0.10) | 41.96 | 1 | **47.43** (0.20) |
>
> Table 2. Comparison of different convolution methods with VGG-19.
> | Dataset | Ori | Dilation | Deformable | Circle | LPSC |
> | :------| :----| :----| :----| :----| :----|
> | # Params (M) | 20.04 | 20.08 | 20.04 | 20.08 | 20.08 |
> | CIFAR-10 | 93.54 (0.06) | 93.46 (0.14) | 92.53 | 93.85 (0.08) | **93.92** (0.06) |
> | CIFAR-100 | 72.41 (0.17) | 73.03 (0.34) | 69.32 | 72.76 (0.28) | **73.13** (0.12) |
>
> Table 3. Comparison of different convolution methods with ResNet-20.
> | Dataset | Ori | Dilation | Deformable | Circle | LPSC | LPSC-CC |
> | :------| :----| :----| :----| :----| :----| :----|
> | \# Params (M) | 0.27 | 0.27 | 0.27 | 0.27 | 0.27 | 0.27 |
> | CIFAR-10 | 91.66 (0.13) | 91.44 (0.10) | 90.27 | 91.76 (0.14) | **91.81** (0.21) | **92.01** (0.08) |
> | CIFAR-100 | 67.56 (0.27) | 66.90 (0.25) | 65.44 | 67.58 (0.28) | **67.63** (0.27) | **68.09** (0.27) |
>
> [A] Kun He, et al. Integrating large circular kernels into cnns through neural architecture search. arXiv:2107.02451, 2021.

---

> > ### Author Response · Authors · 2022-08-02
> > **Added comparisons with more recent methods, explanation on the regularization effect, added image statistics for justification, and added ablation studies (2/3)**
> >
> > **Q2:** Weakness 2: The method does not seem to bring a huge gain, but result into slower training and inference and additional hyperparameters. While this should still be ok for data-limited setup because of the potential regularization effect of the proposed system, it is not clear whether it is better to use the proposed method or just a somewhat larger CNN to achieve similar gains. Please study the regularization / generalization effect of the proposed method further.
> >
> > **A2:** We respectfully remind the reviewer that the gain is achieved without increasing the number of learnable parameters to increase LRF. However, directly using larger conventional convolution kernels will increase the number of parameters greatly. Our LPSC improves different convolutional networks on different datasets in different tasks, ranging from AlexNet to DeepLabV3+, from classification to segmentation. The improvements are obtained without changing any training hyperparameters (learning rate, etc) based on the average results of multiple runs. LPSC even achieves significant improvements in some cases. Moreover, compared with conventional convolution, LPSC has a larger LRF with comparable parameters, or has much fewer parameters with comparable LRF. We believe that these results indeed demonstrate that LPSC is an effective and versatile alternative or supplement to conventional convolution for increasing the LRF.
> >
> > Our LPSC imposes regularization effects in two aspects. 1. It reduces the number of parameters in large convolution kernels and thus reduces the risk of overfitting. In AlexNet in Tab.1, where large kernels are used in the first two layers, LPSC achieves better results with fewer parameters, and the improvement is more significant on the more difficult CIFAR-100 dataset. This indicates that LPSC can better generalize to the test set. 2. LPSC performs average pooling in each region, which actually weakens the weights of faraway regions as shown in Eq. (3), i.e,, the weight $W(l,m)$ for pixels in the $bin(l,m)$ is divided by $N_{l,m}$. In Tab. 7 in Appendix A.2, we evaluate the effects of such weight regularization in Eq. (3) based on AlexNet. “Sum” shows the results by using sum pooling instead of mean pooling in log-polar space pooling in the first two LPSC layers. This is equivalent to removing the regularization and the performances are severely degraded. This is because far regions are exponentially larger than nearer regions. If positions in all regions are treated equally, even if the weight for a far region is not too large, the accumulation of less relevant distant pixels will still produce an overwhelming response. We also try max pooling. It also performs worse than mean pooling. Due to the large LRF, regions with large distance levels for many adjacent center locations will have large overlaps. Some large responses may dominate repeatedly in many regions for different center locations, which suppress other useful local information without the regularization of LPSC. These results have been included in Appendix A.2 in Page 15-16 of the revised manuscript. Thank you for this suggestion.
> >
> > Table 4. Effects of the weight regularization.
> > | Method | Sum | Max | Mean |
> > | :------| :----| :----| :----|
> > | CIFAR-10 | 21.61 | 76.65 | **78.28** |
> > | CIFAR-100 | 5.53 | 44.63 | **47.31** |

---

> > > ### Author Response · Authors · 2022-08-02
> > > **Added comparisons with more recent methods, explanation on the regularization effect, added image statistics for justification, and added ablation studies (3/3)**
> > >
> > > **Q3:** Weakness 3: The motivation of the log-polar space convolution would be better if natural image statistics were used to justify it. Now it stands out as rather ad-hoc method.
> > >
> > > **A3:** Thank you for this suggestion. We have calculated the correlations between image pixels in different log-polar regions and the center pixels on the training set of CIFAR-100. Specifically, for each pixel in each image, we divide its $11 \\times 11$ neighboring area into different regions by LPSC with 3 distance levels, 8 direction levels, and a growth rate of 2. The center pixels of all areas form the center set. The pixels at the same position of all areas also form a pixel set. For each position, we calculate the correlation score between the corresponding pixel set and the center set. The correlation scores of positions in the same region of all training images are averaged to obtain the correlation score between the region and the center pixel. In this way, we obtain the correlation scores from all $3 \\times 8$ regions to the center pixel, as shown in the table below. In further regions, the average correlation of pixels is lower, and the average correlations within different regions of the same distance level are comparable. The farther the region is from the center point, the lower the correlation, and the decay is rapid (approximately exponentially decreasing). These statistics verify the motivation of the log-polar space convolution. We have added this motivation in Section 1 and the statistics results in Appendix A.1.
> > >
> > > Table 5. Correlation scores between different regions and the center pixel.
> > > | Direction level | 4 | 3 | 2 | 1 | 5 | 6 | 7 | 8 |
> > > | :------| :----| :----| :----| :----| :----| :----| :----| :----|
> > > | Distance level 1 | 0.6795 | 0.8268 | 0.6794 | 0.9162 | 0.8323 | 0.6799 | 0.8277 | 0.6800 |
> > > | Distance level 2 | 0.4663 | 0.5844 | 0.4620 | 0.5967 | 0.5967 | 0.4620 | 0.5852 | 0.4665 |
> > > | Distance level 3 | 0.2113 | 0.2779 | 0.1992 | 0.2983 | 0.2976 | 0.1968 | 0.2764 | 0.2097 |
> > >
> > >
> > > **Q4:** Question 1: Although only a bit earlier work, the authors should discuss the relation to this paper : He, Kun, et al. "Integrating Large Circular Kernels into CNNs through Neural Architecture Search." arXiv preprint arXiv:2107.02451 (2021). That work also cites other papers that should be looked at (some of these are already cited by the current paper): “... the deformable convolution (Dai et al., 2017; Zhu et al., 2019) … Similarly, the deformable kernel (Gao et al., 2020) …including quasi-hexagonal convolution (Sun et al., 2016), blind-spot convolution (Krull et al., 2019), asymmetric convolution (Ding et al., 2019), etc.”
> > >
> > > **A4:** Please refer to A1.
> > >
> > > **Q5:** Question 2: Have the authors studied how much performance comes from the polar coordinates and how much from the growing size of areas that are pooled (further areas are pooled more). Could the pooling be done in the normal rectangular grid and how much would be gained by that.
> > >
> > > **A5:** Thank you for this suggestion. We have added comparisons with [A] which actually employs the polar coordinates, results are shown in Appendix A.3, Tab. 9-Tab. 10. Our LPSC outperforms [A], which demonstrates the effect of the exponential increase of outer regions and the pooling-based regularization, as discussed in A2. For normal rectangular grids, we averagely divide the kernel into different square regions, e.g., a $9 \\times 9$ kernel can be divided into $3 \\times 3$ regions with a size of $3 \\times 3$. All positions in the same square region share the same parameter. We denote this alternative convolution method by square convolution, which also increases LRF with fewer parameters. Comparisons with square convolution are shown as follows, where we perform square convolution once since its performance is lower than other methods by a margin. Our LPSC also outperforms the average square convolution with different hyper-parameters, as presented in Appendix A.3. This indicates that the spatial structure designed in log-polar space can better capture contextual information. These results and discussions are included in Tab. 9 and Tab. 10 in Appendix A.3.
> > >
> > > Table 6. Comparison of different convolution methods with AlexNet.
> > > | Dataset | Square | LPSC |
> > > | :------| :----| :----|
> > > | CIFAR-10 | 76.10 | 78.44 (0.12) |
> > > | CIFAR-100 | 44.50 | 47.43 (0.20) |
> > >
> > > Table 7. Comparison of different convolution methods with VGG-19.
> > > | Dataset | Square | LPSC |
> > > | :------| :----| :----|
> > > | CIFAR-10 | 90.01 | 93.92 (0.06) |
> > > | CIFAR-100 | 66.64 | 73.13 (0.12) |
> > >
> > > Table 8. Comparison of different convolution methods with ResNet-20.
> > > | Dataset | Square | LPSC |
> > > | :------| :----| :----|
> > > | CIFAR-10 | 89.84 | 91.81 (0.21) |
> > > | CIFAR-100 | 63.54 | 67.63 (0.27) |

---

> > > > ### Author Response · Authors · 2022-08-08
> > > > **Follow-up on the responses and revised manuscript**
> > > >
> > > > Dear Reviewer aXhX,
> > > >
> > > > Thank you again for your valuable comments. Since the discussion stage is closing soon, we would be grateful if you could let us know whether our responses and revised manuscript have addressed your concerns and whether there are further comments.
> > > >
> > > > Sincerely,
> > > > Authors

---

> > > > > ### Comment · Reviewer_aXhX · 2022-08-08
> > > > > **Thank you**
> > > > >
> > > > > Thank you for your comments! I have updated the review and the score.

---

> > > > > > ### Author Response · Authors · 2022-08-09
> > > > > > **Thanks again**
> > > > > >
> > > > > > Thank you very much again for your valuable and constructive comments, which we believe have improved our manuscript significantly.

---

### Comment · Area_Chair_5Xx9 · 2022-08-08
**After the rebuttal**

Dear Reviewers,

Please read the rebuttal and other reviews. It is now a good moment to discuss (publicly and privately) the paper and reflect on the authors' responses.

Please keep in mind that the discussion period ends on Aug 09 '22 08:00 PM UTC.

Best.
Area Chair

---

### Meta-Review · Area_Chair_5Xx9 · 2022-08-25

**Recommendation:** Accept
**Confidence:** Less certain

**Metareview:**

Thank you for the submission. After reading the paper (twice) and all reviews, my summary could be found below.

Overall, the reviewers agree that the paper has some positive aspects, namely:
+ The idea of log-polar convolution makes intuitive sense since the rectangular pixel quantization is more an artifact of the capture and storage systems rather than a feature of nature.
+ A practical pooling method for easily incorporating the method into existing SW, HW, and networks.
+ The proposed approach is generic and applicable to most existing CNNs, and it is easy to implement.
+ The choice of experiments is motivated well.

However, there is mainly one deficiency reported:
- The method seems to bring some gain in performance, but it also results in slower training and inference, and it requires additional hyperparameters.

I would like to thank the authors for their rebuttal, it helped a lot to improve the paper. I want to also thank the reviewers for their discussions. After the rebuttal, it seems that most concerns have been solved and even though the average score is borderline accept/weak accept, I believe that the paper could be accepted.


**Award:**

No

---

### Decision · Program_Chairs · 2022-09-14

Accept